# Multiple intermolecular interactions facilitate rapid evolution of essential genes

**Huei-Yi Lai, Yen-Hsin Yu, Yu-Ting Jhou, Chia-Wei Liao ⓘ & Jun-Yi Leu ⓘ ✉**

Essential genes are commonly assumed to function in basic cellular processes and to change slowly. However, it remains unclear whether all essential genes are similarly conserved or if their evolutionary rates can be accelerated by specific factors. To address these questions, we replaced 86 essential genes of *Saccharomyces cerevisiae* with orthologues from four other species that diverged from *S. cerevisiae* about 50, 100, 270 and 420 Myr ago. We identify a group of fast-evolving genes that often encode subunits of large protein complexes, including anaphase-promoting complex/cyclosome (APC/C). Incompatibility of fast-evolving genes is rescued by simultaneously replacing interacting components, suggesting it is caused by protein co-evolution. Detailed investigation of APC/C further revealed that co-evolution involves not only primary interacting proteins but also secondary ones, suggesting the evolutionary impact of epistasis. Multiple intermolecular interactions in protein complexes may provide a microenvironment facilitating rapid evolution of their subunits.

Essential genes are a group of genes required for cell survival in normal growth conditions. A broad spectrum of species shares a large proportion of their essential gene repertoires[1,2]. In humans, many genetic diseases have been linked to mutations in essential genes[3]. Therefore, understanding how essential genes change over the course of evolution is important to both basic and medical sciences. Essential genes are mainly involved in core cellular functions that arose early in biotic evolutionary history, so they are commonly assumed to be under strong purifying selection to maintain their physiological functions[4]. This assumption is supported by a genome-wide analysis of gene essentiality between two distantly related yeast species, *Schizosaccharomyces pombe* and *Saccharomyces cerevisiae*, which revealed a high degree of conservation among essential genes despite ~420 Myr of divergence[2]. However, in a recent survey of 414 yeast essential gene mutants, only less than half of them could be complemented by human orthologues[5]. Using a similar approach, 31 of 51 tested bacterial orthologues could substitute for yeast essential genes[6]. Obviously, even though essential physiological functions appear conserved, the underlying molecular or regulatory mechanisms vary sufficiently to result in genetic incompatibility between species.

How can a gene maintain its physiological function and yet change its molecular properties to a level causing genetic incompatibility?

More than 100 years ago, William Bateson introduced the concept of 'epistasis' to describe the puzzling phenotypic variation observed when the same genetic locus was combined with different genetic backgrounds[7]. According to the hypothesis of epistasis, how a gene behaves (and is selected) is determined by the interaction between the gene and its current genetic background. For example, a 'disease-causing' mutation may or may not result in pathogenesis, depending on whether compensatory mutations exist in the genetic background[8]. Non-additive interactions under epistasis between two mutations can lead to phenotypes weaker or stronger than expected according to summation of both individual mutation effects[9]. Thus, theoretically, a conserved physiological function can result from a combination of epistatically interacting non-neutral mutations at the molecular level. However, when the genetic architecture is changed, these mutations may reveal deleterious effects on cell fitness, as observed in orthologue incompatibility between species[10]. Moreover, it raises an intriguing hypothesis that a gene with more genetic interactions may be able to accumulate a broader spectrum of mutations while maintaining a conserved function.

Although epistasis has long been used by geneticists and evolutionary biologists to describe complicated interactions between genetic loci, it is not until recently that biologists started to gain insight

Institute of Molecular Biology, Academia Sinica, Taipei, Taiwan. ✉e-mail: jleu@imb.sinica.edu.tw

into its underlying molecular mechanisms. Epistasis within molecules often involves mutations affecting the conformation or stability of a protein or RNA[11,12]. Moreover, it has been shown to impact mutation fixation and therefore the evolutionary path of proteins[12–14]. However, the mechanisms of epistasis that operate between molecules appear to be more complex and remain largely unresolved. Redundancy of genes or pathways, network topology, physical interactions of molecules and molecular chaperons have all been suggested to play a role in intermolecular epistasis[15]. Nonetheless, it is not clear how each mechanism contributes to long-term evolution, and even less so for higher-order epistasis that involves multiple interactions[16]. As epistatically interacting genetic loci have been observed or widely implicated in quantitative complex traits[17], hybrid vigour, speciation[18,19] and the evolution of sex[20], detailed characterization of intermolecular epistasis would provide a better understanding of various fundamental aspects of evolutionary biology, including the evolution of essential genes.

To know how essential genes diverge between different species and whether there are specific patterns during essential gene evolution, we systematically replaced 86 essential genes in *S. cerevisiae* with corresponding orthologues from species that had diverged from it over different timeframes. Our results show that essential genes exhibit a wide range of variation in their evolutionary trajectories. By analysing the orthologous genes that have quickly become incompatible with the *S. cerevisiae* background, we discovered that intermolecular epistasis plays a key role in their evolution and that conserved physiological functions are maintained by co-evolution of interacting components. Finally, we investigated the fast-evolving anaphase-promoting complex/cyclosome (APC/C) to illustrate how multiple interactions between different components in a large protein complex have influenced the evolutionary pattern of essential genes.

## Broad evolutionary trajectory variation in essential genes

To examine how essential genes change during the course of evolution, we replaced essential genes in *S. cerevisiae* with orthologues from other yeast species (Fig. 1a and Extended Data Fig. 1; see Methods for details). If an orthologue could not rescue cell viability in an essential gene-deleted *S. cerevisiae* mutant, it indicates that the orthologous essential gene has changed to such an extent that it is no longer compatible with the *S. cerevisiae* genomic background. We tested orthologous essential genes from four ascomycete yeast species—*Naumovozyma castellii* (Ncas), *Kluyveromyces lactis* (Klac), *Yarrowia lipolytica* (Ylip) and *S. pombe* (Spom)—that are estimated to have diverged from their common ancestor with *S. cerevisiae* (Scer) about 50, 100, 270 and 420 Myr ago (Ma), respectively[21,22]. By covering a range of species, we anticipated revealing the evolutionary patterns of essential genes.

In *S. cerevisiae*, about 1,000 genes (18% of the genome) are known to be essential for cell viability of the lab strain in rich medium[1]. We selected 86 essential genes involved in various cellular functions for a compatibility test (Supplementary Table 1). Candidate genes with high and low sequence divergence were slightly overrepresented because we speculated that these two types of gene might reveal different patterns at the level of protein function (Extended Data Fig. 2 and Supplementary Table 2). In this study, we only focused on the evolution of protein coding regions because it is an experimentally underexplored topic. In addition, previous studies and our pilot experiments have shown that the regulation of gene expression could change quickly even among closely related species (Supplementary Table 3)[23,24]. Therefore, in our initial screen, we used the Tet-Off promoter to drive both *S. cerevisiae* and orthologous genes, and compared their phenotypes (see Methods). In later experiments, we also used the endogenous promoters from *S. cerevisiae* to confirm that observed incompatibility was not caused by unbalanced stoichiometry.

When growing in rich medium, only 9% of the control strains in which an essential gene was replaced by the Tet–Scer coding sequence

(CDS) copy showed more than a 20% change in growth rate, indicating that replacing the endogenous promoter of an essential gene with the Tet-Off promoter does not impose a heavy burden on the cell in most strains (Extended Data Fig. 3a and Supplementary Table 4). In contrast, when the orthologues from different species were examined in *S. cerevisiae* essential gene-deleted mutants, around 12 to 39% of them failed to rescue cell viability, with the orthologues from more distal donors having lower percentages of compatibility (Fig. 1b).

To further analyse the specific features of different essential genes, we classified them into four types (Supplementary Table 1 and Fig. 1a,c). The majority of tested genes (44/84 = 52%) belong to the 'static' type in which all four orthologues can function in the *S. cerevisiae* background, suggesting that these genes have retained similar molecular functions and interactions for almost half a billion years (Fig. 1c). A recent study also showed that 47% of tested human orthologues were able to substitute for essential genes in yeast cells[5]. Our results evidence that a large proportion of essential genes evolve slowly and are conserved across divergent lineages. The others probably reflect lineage-specific evolution. The 'gradual' type comprises 25 genes (25/84 = 29%) that have evolved incompatibility over the course of yeast evolution, and the incompatibility appears consistently through more distant species (including the ones that only show incompatibility in the *S. pombe* orthologues). Genes of the 'punctuate' type (7/84 = 8%) show branch-specific incompatibility not in accordance with the pattern of species divergence, perhaps reflecting specific changes to individual lineages (Fig. 1c). For example, *S. cerevisiae* and *Y. lipolytica* have different tRNA gene repertoires[25], and the incompatibility of several Ylip orthologues may partly be explained by compromised translation due to differences in codon usage. In the 'fast' type (8/84 = 10%), none of the orthologues are compatible with the *S. cerevisiae* genome. To rule out the possibility that the observed incompatibility was due to unbalanced stoichiometry caused by the Tet-Off promoter, we expressed the 'fast'-type genes of *N. castellii* under the endogenous promoters from *S. cerevisiae*. None of these constructs rescued the viability (Supplementary Table 3), indicating that incompatibility resulted from the changes in CDSs. These data suggest that either these genes are fast evolving and become incompatible in species as closely related as *N. castellii* and *S. cerevisiae* (that is, within 50 Myr), or the *S. cerevisiae* genome has accumulated branch-specific mutations that make other orthologues incompatible. Although our experiments could not distinguish these two possibilities, our following sequence analyses revealed that the fast-type genes also change their sequences quickly in other lineages, suggesting their general fast-evolving nature.

We compared the protein sequence identity and non-synonymous substitution rate (Ka) between these four types of essential gene to see whether the compatibility patterns are correlated with sequence divergence. Indeed, the fast group has the highest evolutionary rate and protein sequence divergence, and the static group has the lowest ones (Fig. 1d and Extended Data Fig. 4). To test whether fast evolution only occurs in the *S. cerevisiae* lineage, we calculated the non-synonymous substitution rate of tested orthologues from all species pairs (Supplementary Table 5) and examined the correlation of Ka between different species pairs. If the same group of genes always behave similarly in different lineages, we expected to see high correlations between them. In contrast, if the pattern is only specific to the *S. cerevisiae* lineage (for example, fast-evolving genes only change quickly in *S. cerevisiae*, but not in other species), we expected to see much lower correlations between *S. cerevisiae*-containing pairs and non-*S. cerevisiae* pairs. High correlations were observed in all pairwise comparisons (Spearman's correlation coefficient $\rho = 0.85$–$0.99$, $P = 2.09 \times 10^{-25}$–$2.66 \times 10^{-69}$, Spearman's rank correlation; Fig. 1e and Supplementary Table 6), suggesting that most tested genes maintain consistent evolutionary characteristics in all five species.

Next, we examined growth rates of the viable strains carrying the compatible orthologues (Extended Data Fig. 3b,c and Supplementary

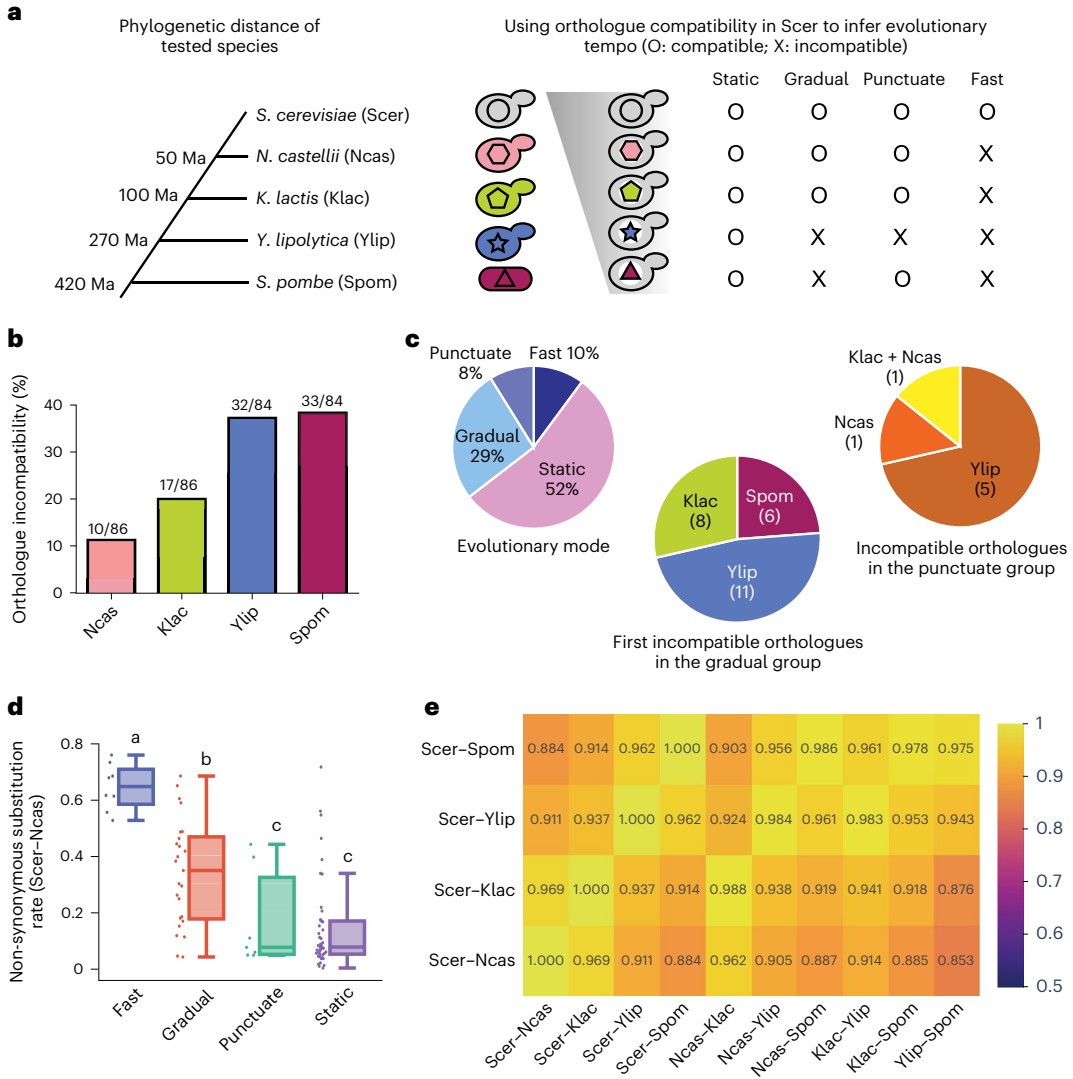

**Fig. 1 | Functional assays reveal incompatibility between essential gene orthologues and the *S. cerevisiae* background. a**, Orthologues from four additional Ascomycota species were used to study the evolution of essential genes. These four species diverged from *S. cerevisiae* over different timeframes, allowing us to characterize when the orthologues had become incompatible. To measure compatibility, *S. cerevisiae* essential genes were replaced by orthologues from the other four species and then cell viability was examined. Depending on the pattern of compatibility between orthologues, we classified the tested essential genes into four types: static, gradual, punctuate and fast. **b**, A total of 12 to 39% of tested orthologues are incompatible with the *S. cerevisiae* background. We performed a chi-square test to examine whether some species have more incompatible orthologues than the others and incompatible gene numbers are significantly different between different species (chi-square test, $n = 340$, $P = 2.2 \times 10^{-5}$). **c**, Distributions of essential genes exhibiting different evolutionary patterns. Only genes for which all four orthologues were tested are included in this analysis. Left: percentage of genes with different evolutionary patterns. Middle: the species when first incompatible orthologues were observed in the gradual group. Right: species-specific incompatibility in the punctuate group. Numbers of genes are shown in parentheses. **d**, The fast-type genes have the highest evolutionary rates between *S. cerevisiae* and *N. castellii*. Boxplots indicate median (middle line), 25th and 75th percentile (box), and minimum and maximum (whiskers). Distributions with different letters (above each boxplot) are significantly different from each other (fast: $n = 8$; gradual: $n = 25$; punctuate: $n = 7$; static: $n = 44$, two-sided Mann–Whitney $U$ test, $P$ values <0.05). See also the Source data for detailed statistical information. **e**, Most tested genes exhibit consistent evolutionary characteristics in all five species. Non-synonymous substitution rates (Ka) of tested orthologues from all species pairs were calculated (Supplementary Table 5) and high correlations of Ka between different species pairs were observed (one-tailed Spearman's rank correlation, $\rho = 0.85$–$0.99$, $P = 9.29 \times 10^{-24}$–$6.10 \times 10^{-66}$; see Supplementary Table 6 for all $P$ values). Spearman's correlation coefficients are shown in the figure.

Table 4). *S. cerevisiae* strains carrying *N. castellii* orthologues showed similar growth rates to those hosting the control Tet–Scer copy (paired *t*-test, $n = 63$, $P = 0.347$). In contrast, orthologues from more distal donors resulted in lower growth rates (paired *t*-test, Scer versus Klac, $n = 57$, $P = 0.026$; Scer versus Ylip, $n = 42$, $P = 0.003$; Scer versus Spom, $n = 41$, $P = 0.009$; Extended Data Fig. 3b). These results indicate that many subtle differences had gradually accumulated in these distant orthologues even though they remain compatible with the *S. cerevisiae* genome.

## Interactor co-expression rescues the incompatible orthologues

The observed orthologue incompatibility could result from changes in the function or interactions. As the primary functions of tested genes are essential for cell viability, a major functional switch is unlikely and only subtle alterations may occur. On the other hand, we found that the proteins encoded by the 'fast'-type genes all work as functional subunits of large stable protein complexes[26], including Apc1, Apc2, Apc4 and Apc5 in the APC/C[27], Sec20 in the soluble N-ethylmaleimide-sensitive

factor (NSF) attachment protein receptor (SNARE) complex[28], Taf8 in the transcription factor II D (TFIID) complex[29], Spp382 in the spliceosome disassembly complex, and Cdc13 in the telomere maintenance Ctc1–Stn1–Ten1 (CST) complex[30] (Supplementary Tables 1 and 7). This finding raises the possibility that phenotypic stasis may be achieved by co-evolution of interacting proteins (that is, changes in the target genes followed by complementary mutations occurring on interacting partners), leading to changes in molecular structure but not physiological function.

Under the protein co-evolution hypothesis, we expected the incompatibility of fast-evolving orthologues to be rescued by co-expression of their interacting partners from the same species. We tested this hypothesis using incompatible orthologues from *N. castellii* because this is the most closely related species to *S. cerevisiae* among those we tested and molecular changes are probably more complex in more divergent species. We selected all possible interacting partners of these incompatible genes based on published biochemical experiments (Supplementary Table 7). In six out of the eight fast-type genes, we were able to uncover compensatory partners; the incompatibilities of Ncas–Apc2, –Apc4, –Apc5, –Spp382, –Sec20 and –Taf8 were rescued by co-expressing one of their direct interacting partners, that is, Ncas–Apc11, –Apc5, –Apc4, –Ntr2, –Tip20 and –Taf10, respectively (Supplementary Table 7). The only exceptions were Ncas–Cdc13 and Ncas–Apc1. Cdc13 is a telomere binding protein, so it is possible that the incompatibility could have arisen from a mismatch between the protein and DNA motifs rather than protein–protein interactions[30]. Co-expressing other individual Ncas–APC/C essential components did not rescue the incompatibility of Ncas–Apc1. We speculate that it may be necessary to co-express multiple interacting partners to rescue this case of incompatibility because Apc1 is a scaffold protein that bridges different subunits of the APC/C complex.

To ensure that the observed APC/C subunit incompatibility and protein co-evolution are not a phenomenon specific to the *S. cerevisiae* lineage, we also tested the compatibility of Scer–Apc2 in the *N. castellii* background. Scer–Apc2 alone failed to rescue the viability of an *N. castellii apc2* deletion mutant. However, the mutant cells became viable if we co-expressed *Scer–APC2* and *Scer–APC11* (Supplementary Table 8 and Methods). These results provide strong evidence that co-evolution of interacting proteins contributes substantially to the observed incompatibility, especially for the fast-type genes.

## Multiple domains are involved in Ncas–Apc2 incompatibility

Next, we aimed to understand the molecular processes of the co-evolution of an essential gene and its partner. Two co-evolving pairs, Apc2–Apc11 and Apc4–Apc5, are components of the APC/C complex, which may serve as an interesting case on how intermolecular interactions influence the evolution of protein subunits in a large protein complex. APC/C regulates a variety of important cellular processes such as cell division, differentiation, genome stability, autophagy and cell death, as well as being linked to carcinogenesis[31]. APC/C is a large protein complex in which 9 of 13 major components are essential for *S. cerevisiae* cell viability. One co-evolving pair, Apc4 and Apc5, are the scaffold proteins of APC/C and the other pair, Apc2 and Apc11, are the enzymatic core of the E3 ubiquitin ligase, APC/C, which controls cell cycle progression by ubiquitinating cell cycle regulators[32]. Protein structural information is available for Apc2 and Apc11, as well as other APC/C components, making this a good paradigm for studying the evolution of intermolecular interactions in detail[27,33].

Apc2 contains two protein-binding domains[27,34]. Previous biochemical and cryogenic electron microscopy structural data indicate that the amino (N)-terminal cullin domain (NTD) binds the scaffold protein Apc1 and the carboxy (C)-terminal globular domain (CTD) interacts with Apc11 and Doc1, which helps the complex to position its substrates and activators (Fig. 2a and Extended Data Fig. 5)[34]. We constructed two

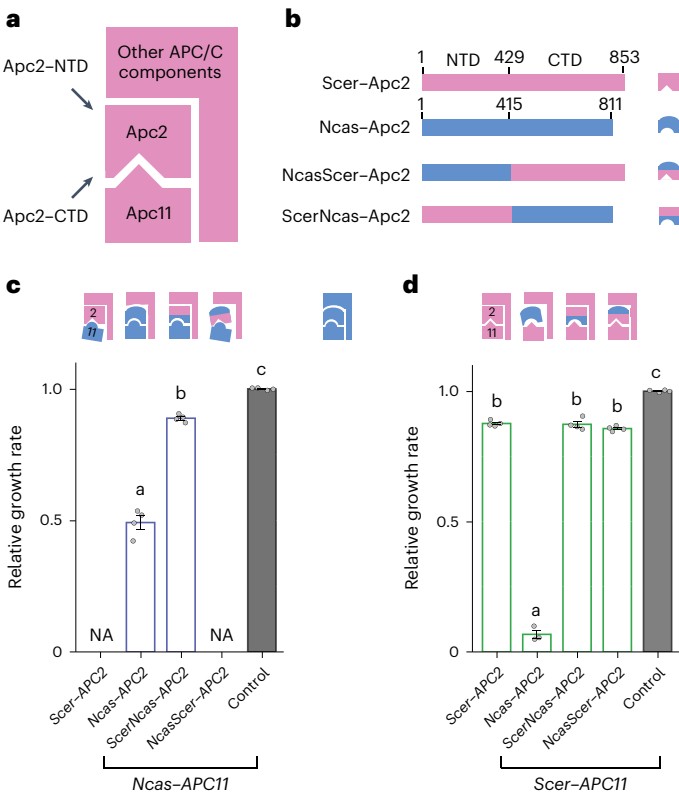

**Fig. 2 | Both protein-binding domains of Apc2 contribute to incompatibility. a**, A simplified diagram depicting the interactions between Apc2, Apc11 and other APC/C components, which we employ in subsequent figures to illustrate how the structure of APC/C is affected by substituted orthologous proteins. The CTD of Apc2 is the main interacting domain with Apc11 and the NTD of Apc2 interacts with other APC/C components. **b**, Construction of chimeric proteins of Scer–Apc2 and Ncas–Apc2. *N. castellii* (Ncas) proteins are in blue and *S. cerevisiae* (Scer) proteins are in pink. **c**, Cells with the NTD of Scer–Apc2 or Ncas–Apc2 exhibit different fitness, suggesting that the NTD of Apc2 also contributes to the stability of the whole complex. The blue complex at top right represents the Ncas–APC/C. NA, data not available due to cell inviability. Control, the *apc2Δ apc11Δ* shuffling strain carrying the plasmid containing *Scer–APC2* and *Scer–APC11*. **d**, Cells carrying ScerNcas–Apc2 and Scer–Apc11 do not have obvious growth defects, suggesting that the incompatibility of Ncas–Apc2 is not completely due to mis-interaction between the CTD of Ncas–Apc2 and Scer–Apc11. All compatibility tests of different proteins were performed in an *apc2Δ apc11Δ* shuffling strain hosting plasmids carrying the orthologous or chimeric genes being driven by the Tet-Off promoter. Growth rates were normalized to that of a control strain (*apc2Δ apc11Δ* + *Scer–APC2 Scer–APC11*). Error bars represent the standard error of the mean from four independent colonies. Columns sharing the same letters are not significantly different ($a = 0.05$, Tukey-adjusted $P$ values). Source data and detailed statistical information are provided in the Source data. See also Extended Data Fig. 5.

chimeric proteins, NcasScer–Apc2 and ScerNcas–Apc2, in which the NTDs of Scer–Apc2 and Ncas–Apc2 were swapped (Fig. 2b). We then tested the functionality of these chimeric proteins in the presence of Ncas–Apc11 (Fig. 2c) or Scer–Apc11 (Fig. 2d) to assess the contributions of the different domains to subunit incompatibility. Our results show that NcasScer–Apc2 could not function with Ncas–Apc11 and further suggested that the mis-interaction with Scer–Apc2 contributes to the incompatibility of Ncas–Apc11 (Fig. 2c).

However, cells carrying ScerNcas–Apc2 were viable when it was co-expressed with Scer–Apc11 (Fig. 2d), suggesting that the Ncas CTD alone is not sufficient to cause Ncas–Apc2 incompatibility. One possible explanation is that even though the Ncas CTD of ScerNcas–Apc2 has difficulties in interacting with Scer–Apc11 (Fig. 2d), the

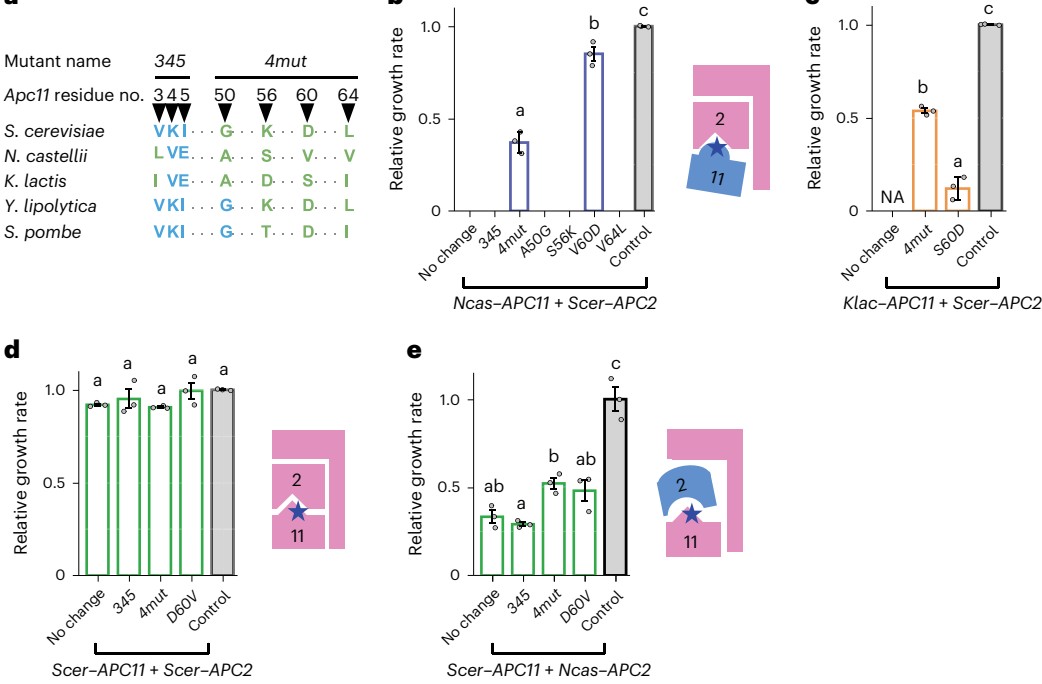

**Fig. 3 | A few diverged residues play a critical role in Apc11 orthologue incompatibility. a**, Seven residues in conserved domains of Apc11 were found to exhibit different patterns between compatible and incompatible orthologues. The amino acids shared by incompatible orthologues are in pink and those shared by compatible orthologues are in blue. Amino acids deviating from the majority of aligned orthologues are in green (Extended Data Fig. 7). **b**, Replacing residue 60 in Ncas–Apc11 rescues incompatibility with Scer–Apc2. Individual or multiple mutations were introduced into *Ncas–APC11* to test for compatibility with Scer–Apc2. Only *V60D* and *4mut* (quadruple substitution of residues 50, 56, 60 and 64) exhibited detectable rescue effects. Control, the *apc2Δ apc11Δ* shuffling strain carrying the plasmid containing *Scer–APC2* and *Scer–APC11*. **c**, Replacing residue 60 in Klac–Apc11 only partially rescues incompatibility with Scer–Apc2,

whereas *Klac–APC11–4mut* exhibits higher compatibility. **d,e**, Introduction of *N. castellii* residues into Scer–Apc11 does not result in obvious incompatibility with Scer–Apc2 (**d**). Nonetheless, these mutations increase the compatibility of Scer–Apc11 with Ncas–Apc2 (**e**), suggesting that residue 60 is crucial for the interaction between Apc11 and Apc2. All *APC11* mutants and *APC2* were driven by the Tet-Off promoter. Growth rates were normalized to that of the control strain. NA, data not available due to cell inviability. Error bars represent the standard error of the mean from three independent colonies. Columns sharing the same letters are not significantly different (*a* = 0.05, Tukey-adjusted *P* values). Source data and detailed statistical information are provided in the Source data. Blue stars indicate the substituted residues in the interacting interface. No change means the template is the wild-type Ncas–Apc11 or Scer–Apc11.

interactions between the Scer NTD of this chimeric protein and the other *S. cerevisiae* APC/C components may stabilize the complex structure. This hypothesis is supported by our observation that in the presence of Ncas–Apc11 (Fig. 2c), cells hosting ScerNcas–Apc2 grew better than Ncas–Apc2-hosting cells, despite the interaction interfaces between Apc11 and Apc2 (that is, Ncas–Apc11 and the CTD of Ncas–Apc2) being exactly the same in both cases. However, mis-interaction of the NTD alone is also insufficient to disrupt APC/C function (Fig. 2d). Our results suggest that the individual domains of Apc2 interact with different proteins in the APC/C complex (that is, the CTD of Apc2 with Apc11 and the NTD of Apc2 with other APC/C components) and the effect of each interaction on cell fitness depends on the other ones.

When we tested the chimeric proteins, we observed marginal growth of cells carrying *Ncas–APC2* and *Scer–APC11* (Fig. 2d), in contrast to the complete incompatibility of *Ncas–APC2* observed in our screening experiments. We found that this inconsistency was caused by a high expression level of *Scer–APC11* driven by the TetO₇ promoter we used in this experiment. When we replaced *tet-Scer–APC11* with a *Scer–APC11* carrying an endogenous promoter (as in the screening experiment), complete incompatibility was again observed (Extended Data Fig. 6). In addition, when Scer–Apc11 was expressed from its native promoter, ScerNcas–Apc2 showed a lower growth rate than NcasScer–Apc2, with both chimeric proteins exhibiting lower fitness than Scer–Apc2. These findings support our hypothesis that interactions between the NTD of Apc2 and other APC/C components also contribute to the co-evolution of Apc2 and Apc11.

## A few critical Apc11 residue changes rescue incompatibility

Compared with Apc2 (853 amino acids (a.a.) for Scer–Apc2), Apc11 is a much smaller protein (165 a.a. for Scer–Apc11), and the compatibility of Apc11 orthologues with the *S. cerevisiae* genome shows an interesting punctuate pattern. Whereas orthologues from less-divergent species (Ncas–Apc11 and Klac–Apc11) exhibit incompatibility, orthologues from more divergent species (Ylip–Apc11 and Spom–Apc11) are compatible (Supplementary Table 1). Comparative sequence analysis of Apc11 from different species revealed that several amino acid residues in the conserved regions of *S. cerevisiae* Apc11 (a.a. 3, 4, 5, 50, 56, 60 and 64) are more similar to those of *Y. lipolytica* and *S. pombe* than they are to those of more recently diverged species (Fig. 3a and Extended Data Fig. 7). These residues are located near the interaction interface with Apc2 (Extended Data Fig. 5), suggesting that Ncas–Apc11 incompatibility might result from rapid co-evolution of the Apc2–Apc11 interaction. Hereafter, the nomenclature of these seven residues follows their relative positions in the Scer–Apc11 sequence.

To examine whether these divergent sites really contribute to the difference in compatibility, we substituted combinations of these seven Scer–Apc11 residues into Ncas–Apc11 and vice versa. We then tested these mutants for compatibility with Scer–Apc2. Interestingly, only replacing residue 60 in Ncas–Apc11 with the Scer–Apc11 sequence (V60D) rescued the incompatible interaction of Ncas–Apc11 with Scer–Apc2 (Fig. 3b). Replacement of residue 60 in Klac–Apc11 (S60D) also partially rescued Klac–Apc11 incompatibility with Scer–Apc2 (Fig. 3c).

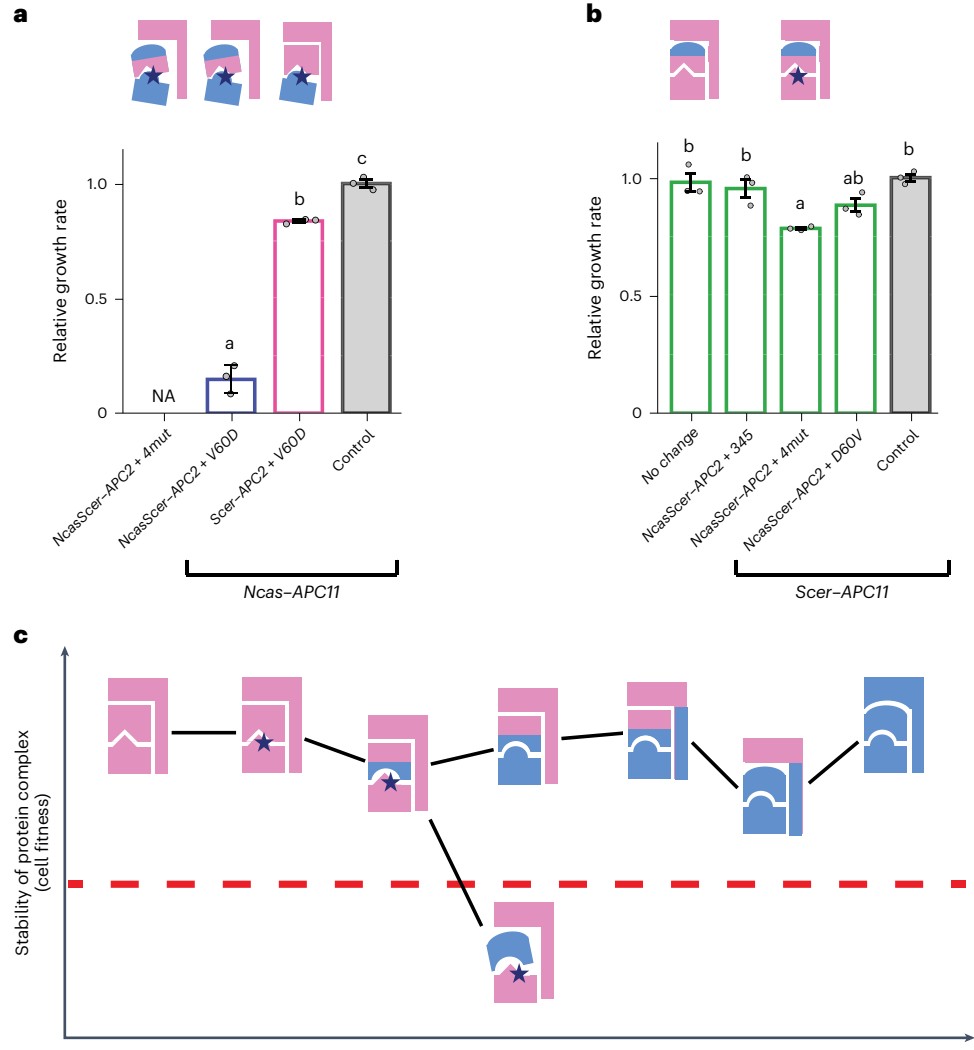

**Fig. 4 | Effects of *APC11* mutants depend on the stability of the protein complex. a**, *Ncas–APC11–V60D* mutants exhibit different fitness in the *NcasScer–APC2* and *Scer–APC2* backgrounds. The Apc2–Apc11 interacting domains remain the same in these two strain backgrounds, but the APC/C complex is less stable in the *NcasScer–APC2* background due to the foreign NTD of NcasScer–Apc2. **b**, *Scer–APC11–4mut* cells exhibit growth defects in the *NcasScer–APC2* background. The *4mut* and *D60V* mutants (see Fig. 3d) show no obvious effect in the *Scer–APC2* background that has a more stable APC/C structure. Growth rates were normalized to that of the control strain. NA, data not available due to cell inviability. Error bars represent the standard error of the mean from three independent colonies. Columns sharing the same letters are not significantly different (*a* = 0.05, Tukey-adjusted *P* values). Source data and detailed statistical information are provided in the Source data. Blue stars indicate the substituted residues in the interacting interface. No change means the template is the wild-type Ncas-Apc11 or Scer-Apc11. **c**, Illustration of the possible evolutionary trajectories of a protein complex under the influence of multiple protein interactions. Mutations accumulated in the interface of different subunits will be cryptically neutral if the structure of the complex is stably maintained by other components. As long as the effect of the mutations in the interface of primary or secondary interacting proteins does not exceed the buffering capacity of the complex (represented by the red dashed line), these mutations may accumulate in the population. Moreover, these mutations may allow interacting partners to further mutate, leading to co-evolution. However, phenotypic variation (or incompatibility) will be revealed if the stability of the complex is compromised sufficiently to exceed the buffering capacity.

Furthermore, an *N. castellii*-like D60V mutant of Scer–Apc11 could interact with Ncas–Apc2 and improved cell growth (Fig. 3e). This D60V Scer–Apc11 mutant still interacted with Scer–Apc2 and did not cause severe growth defects (Fig. 3d), suggesting that interaction stability is not solely determined by residue 60. Together, these results suggest that mutations of residue 60 represent a permissive step allowing the interacting interface of Apc2 and Apc11 to further change.

Even though replacements of the remaining six divergent sites did not rescue Ncas–Apc11 incompatibility, some of them interacted epistatically with residue 60. The Ncas–Apc11–4mut mutant that carried a quadruple replacement (sites 50, 56, 60 and 64) exhibited weaker compatibility with Scer–Apc2 than the Ncas–Apc11–V60D mutant

(Fig. 3b). In contrast, the same quadruple replacement in the Klac–Apc11–4mut mutant showed higher compatibility with Scer–Apc2 (Fig. 3c). These results indicate that Ncas–Apc11 and Klac–Apc11 have probably evolved differently despite sharing similar divergent patterns at those seven amino acid positions.

## Interactions between different APC/C components are crucial

Our earlier chimeric Apc2 experiments revealed that the interaction of the NTD of Apc2 with other APC/C proteins might influence the compatibility between Apc2 and Apc11 orthologues (Fig. 2 and Extended Data Fig. 6). APC/C is a large protein complex and its stability is influenced by

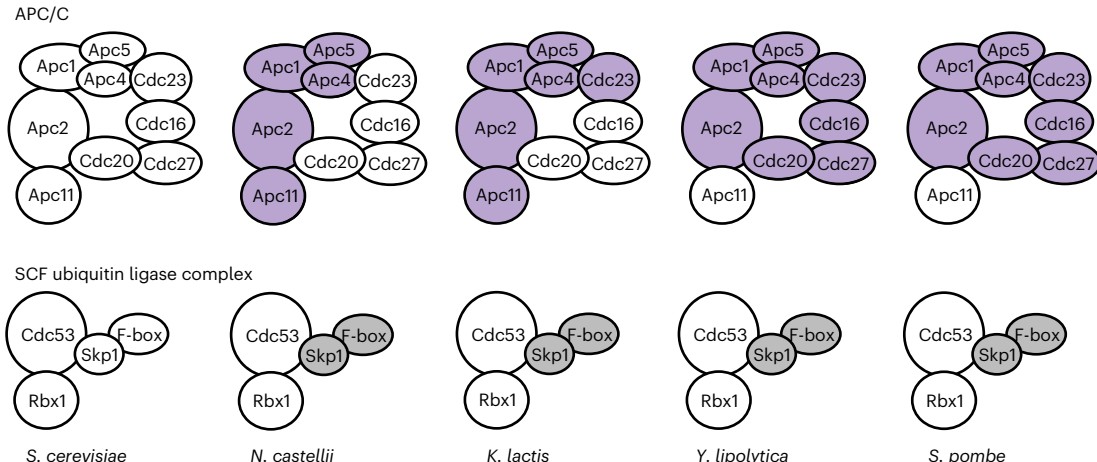

**Fig. 5 | APC/C and its homologue, SCF, exhibit different evolutionary patterns.** Both APC/C and SCF complexes are the ubiquitin E3 ligases involved in cell cycle progression, but APC/C has far more subunits than SFC. Most APC/C core subunits evolve incompatibility between *S. cerevisiae* and *N. castellii*. In contrast, SCF core subunits remain static in all species tested (Supplementary Table 1). Orthologous APC/C subunits incompatible with *S. cerevisiae* are coloured purple and uncharacterized SCF subunits are coloured grey.

multiple interactions between different protein subunits. It suggests that mutations causing a weakened interaction between two subunits may be tolerated (so the complex is still functional) if other interactions of these subunits are intact. However, the same mutations may cause a drastic effect if they occur in an unstable background. To elucidate the possible background effect, we investigated how the compatibility of mutations at the interaction interface of Apc11 with Apc2 is influenced by the interaction of Apc2 with other APC/C subunits.

We examined the fitness of the Ncas–Apc11–V60D substituent in the background of Scer–Apc2 and NcasScer–Apc2, which probably has a compromised interaction with other Scer–APC/C proteins. Figure 4a showed that the compatibility of the Ncas–Apc11–V60D substituent was much reduced in the NcasScer–Apc2 background compared with the Scer–Apc2 background. Similarly, the Ncas–Apc11–4mut mutant completely failed to rescue the viability of NcasScer–Apc2-hosting cells (Fig. 4a), although it did for Scer–Apc2-hosting cells (Figs. 3b and 4a). We also observed a similar dependency when Scer–Apc11 substituents (Scer–Apc11–D60V and Scer–Apc11–4mut) were examined. Previously, no obvious defect was observed between the Scer–Apc11 substituents and Scer–Apc2 (Fig. 3d). However, we found that the compatibility of the Scer–Apc11 substituents was significantly reduced when coexisting with NcasScer–Apc2 (Fig. 4b).

Together, these results show that even though the NTD of Apc2 does not directly interact with Apc11, it can influence the impacts of the mutations affecting the Apc2–Apc11 interaction. It raises a possibility that destabilizing mutations may be sustained and further influence the evolutionary trajectory of the protein complex if they occur in a background in which the structure of the complex is stably maintained (Fig. 4c).

Our experimental assessment of Apc2–Apc11 co-evolution suggests that multiple protein interactions in the APC/C complex may enable it to tolerate changes in its protein interaction interfaces, allowing co-evolution of two of its subunits. To investigate how this scenario influences the evolution of other APC/C subunits, we examined orthologue compatibility for all nine essential genes of the APC/C complex in *S. cerevisiae*. We found that more than half of the orthologues from *N. castellii* exhibited incompatibility, including the core enzymatic proteins Apc2 and Apc11, as well as the structural proteins Apc1, Apc4 and Apc5 (Fig. 5). In the more divergent species *K. lactis*, the orthologue of Cdc23—a subunit of the TPR lobe that directly interacts with Apc5—was also incompatible. For *Y. lipolytica* and *S. pombe*, all tested orthologues except for Apc11 exhibited incompatibility (Fig. 5 and Supplementary Table 1).

Co-expression of Ncas–Apc2 and Ncas–Apc11 as well as Ncas–Apc4 and Ncas–Apc5 can reverse the incompatibility of individual orthologues (Supplementary Table 8). However, we failed to see a similar reversion when co-expressing orthologous pairs (Apc2–Apc11 and Apc4–Apc5) from the more distal donor, *K. lactis* (Supplementary Table 8). These results suggest that, in addition to co-evolution between primary interacting subunits of the *K. lactis* APC/C complex, other incompatible mutations (for example, co-evolution among secondary interacting subunits) have accumulated when species further diverge.

## Subunits of the Skp–Cullin–F-box complex are static

In several cases of fast-evolving genes, the molecular function of the gene products has been implicated in driving their evolution[35,36]. To understand whether the unique evolutionary pattern of APC/C subunits was simply driven by its molecular function, we examined the orthologue compatibility of its homologue, the Skp–Cullin–F-box (SCF) complex. SCF is another E3 ubiquitin ligase involved in cell cycle regulation[34], and it has been suggested that it evolved from the same ancestral protein complex as APC/C[37]. However, the structure of SCF is less complex, consisting of only four proteins in *S. cerevisiae*. We performed orthologue replacements of Cdc53 and Rbx1, which are homologues of Apc2 and Apc11, and found that all orthologues of *CDC53* and *RBX1* (from the same four divergent species tested for the APC/C complex) were compatible with the *S. cerevisiae* genome (Fig. 5 and Supplementary Table 1). Such high conservation greatly contrasts with our findings for Apc2 and Apc11, suggesting that the molecular function is not the primary driving force for the evolutionary pattern of Apc2 and Apc11.

## Discussion

A paradox in evolutionary biology is that some phenotypic traits across species do not seem to be subject to progressive changes over time and instead are shared by divergent modern species or even their fossilized ancestors. Invariant phenotypes or evolutionary stasis is often overlooked in evolutionary studies, which tend to heavily focus on traits that rapidly change in response to environmental or other factors. Moreover, the purifying selection that is widely accepted as responsible for evolutionary stasis further precludes the interests of studying the underlying mechanisms[38]. However, recent discoveries have challenged the conventional perception of phenotypic stasis. For example, a comparison of the vulval development of different nematode species

showed that while a constant phenotype is maintained, quantitative differences in cellular responses exist between closely related *Caenorhabditis* species and qualitative changes in regulatory pathways occur between distantly related nematodes[39]. Similarly, studies of yeast have shown that, for some transcription circuits, the regulators and outputs have been conserved among different species, yet the regulatory networks have been extensively rewired[40].

Evolutionary stasis can be observed at different levels of biological organization. Essential genes represent a good paradigm for studying evolutionary stasis at the gene level. Their protein sequences are more highly conserved[4,41] and they are more likely to have orthologues across diverse organisms[1,2]. Nonetheless, recent systematic studies have revealed that a subset of essential genes can become dispensable if cells evolve bypassing mutations[42,43]. Genome-wide gene deletions from different strains of *S. cerevisiae* have also indicated that some essential genes are strain-specific[44]. These observations suggest that essential genes can possess diverse characteristics. Here, we demonstrate that essential genes can indeed exhibit various evolutionary patterns, with some of them suggesting that complete genetic incompatibility has occurred within 50 Myr of evolution. We further confirmed experimentally that the evolution of incompatibility in this type of essential genes is mainly driven by protein co-evolution and is probably facilitated by multiple protein interactions. This phenomenon is distinct from previously described 'evolvable' essential genes, the essentiality of which can be bypassed by mutations in another gene or pathway[42,43]. In these latter cases, whether or not an essential gene is 'evolvable' greatly depends on its molecular functions and the pathways involved.

Essential cellular functions are strongly constrained by purifying selection because any defect can easily lead to fitness loss or lethality of the organisms. However, selection at the gene level is more complicated. We show that protein co-evolution allows cells to tolerate extensive changes in the interaction interfaces of essential proteins without losing their conserved functions (Supplementary Table 7). Thus, as long as the structure of the entire complex is maintained, the interaction interfaces can be shaped in distinct ways among different lineages. As a consequence, seemingly neutral (or cryptic) mutations can eventually lead to incompatibility between interacting protein complex subunits (Bateson–Dobzhansky–Muller incompatibility) if their link with the arising background is disrupted[18]. Interestingly, when we compared static and non-static groups, non-static genes were more likely to encode protein complex subunits than static genes (34/40 versus 23/44, chi-square test, $P$ = 0.001; Supplementary Table 1)[26].

In addition to facilitating the accumulation of cryptic genetic variation, it has also been theorized that epistasis can shape the evolutionary trajectory of a gene, that is, a gene under the influence of complex epistatic effects may be able to access different fitness peaks more easily and consequently be more evolvable[45–47]. APC/C is a large protein complex that is hypothesized to have been present in the last eukaryotic common ancestor, and most of its subunits can still be found in all present-day eukaryotic lineages with few exceptions[48,49]. However, we found that individual orthologous subunits cannot be exchanged among closely related yeast species, revealing that components of this ancient complex can change in a short period of evolutionary time. Our analysis of APC/C subunits provides a possible scenario for how epistasis can accelerate the evolution of multiple components of an essential protein complex. Moreover, we found the 'fast' group genes all belong to the subunits of large protein complexes in *S. cerevisiae* (the sizes of which are among the top 16% of this yeast's essential protein complexes; Supplementary Table 7). It implies that the complex environment may have an important influence on gene evolvability.

During yeast evolution, a large hybridization event happened between a *K. lactis*-like lineage and a *Zygosaccharomyces rouxii*-like lineage to generate the common ancestor of *N. castellii* and *S. cerevisiae*[50,51].

Following the hybridization event, many duplicated genes were lost and now they only account for less than 20% of modern yeast genomes[52]. It raises a possibility that maybe different duplicated copies of the APC/C components were maintained in *N. castellii* and *S. cerevisiae*, which led to observed incompatibility. However, this hypothesis is not supported by the genome synteny data. If *N. castellii* and *S. cerevisiae* keep different copies of the APC/C genes from the hybrid genome, we expect that flanking regions of the APC/C genes should have different gene contents, because two homeologous chromosomes (one from the *Z. rouxii*-like lineage and another from the *K. lactis*-like lineage) lost different genes after whole-genome duplication. However, the gene synteny data show that similar orthologous gene sets are kept in the regions flanking to APC/C genes (*APC1*, *APC2*, *APC4*, *APC5* and *APC11*) in *N. castellii* and *S. cerevisiae*[52], suggesting that both species retained the same copy of APC/C genes after whole-genome duplication.

Stable integrity of large protein complexes is critical for their functions and this is maintained by the interactions between multiple protein interfaces. These molecular interaction microenvironments can tolerate changes to the interacting interfaces of individual subunits or even interaction partners without causing a loss of the integrity of a protein complex or its physiological functions[53]. However, if the stability of the complex is compromised by further environmental or genetic perturbations, phenotypic variation would be manifested because the effect of accumulated mutations would no longer be buffered. This phenomenon is similar to the 'capacitor' (or 'genetic buffering') model proposed in studies of Hsp90, a protein implicated in facilitating population adaptation to changing environments[54]. One-third of the *S. cerevisiae* proteome has been identified as consisting of protein complex components, and the functions and structures of a large fraction of these complexes remain understudied[26]. Similar proportions of protein complexes are probably present in other eukaryotic organisms[55]. More studies are required to understand the contribution of protein complexes to long-term population adaptation and why some protein complexes can change quickly while others remain static.

## Methods

### Construction of plasmids

Two types of yeast single-copy plasmid were used in the orthologue compatibility test. The pRS416–Scer, pRS413–Ylip and pRS413–Spom plasmids carry the CDS of a *S. cerevisiae*, *Y. lipolytica* or *S. pombe* gene and its regulatory elements, including the upstream 1 kb and downstream 0.5 kb regions (Supplementary Table 9). The pRS416–Scer plasmid was used to supply the essential gene function when we deleted the genomic copy of an essential gene. pRS413–Ylip and pRS413–Spom plasmids were used to test the compatibility of the promoters when the data were compared with those of pRS413–tet–orthologue plasmids. We tested 36 orthologous genes from *Y. lipolytica* and *S. pombe* (Supplementary Table 3). Only 16 of the pRS413–Ylip or pRS413–Spom plasmids (16/36 = 44%) complemented the *S. cerevisiae* mutants, despite the fact that all the pRS413–tet–orthologue plasmids could complement. The pRS413–tet–orthologue plasmid contains a tetracycline responsive element (tetO7 from the pUC*tet*O7 plasmid), a *S. cerevisiae CYC1* terminator and the CDS of a *S. cerevisiae* essential gene or its fungal orthologue. The orthologous CDSs were polymerase chain reaction (PCR)-amplified from the cDNA of four ascomycete yeast species: *N. castellii*, *K. lactis*, *Y. lipolytica* and *S. pombe*. As some fungal genes have multiple introns and may retain these introns in purified mRNA, we obtained the templates by de novo synthesis (GENEWIZ). The pRS413–tet–orthologue plasmids were used to test the compatibility of orthologues. In cases where expression of two plasmids was required, such as in the rescue experiment in Supplementary Table 7, pRS415 was used to generate pRS415–tet–orthologue plasmids.

Chimeric *APC2* genes were constructed by swapping the NTDs of Scer-Apc2 (residues 1–429) and Ncas-Apc2 (residues 1–415). To test the effect of expression levels of *Ncas–APC11*, a plasmid containing

the regulatory elements of *Scer–APC11* and the CDS of *Ncas–APC11* was also constructed. Plasmid construction was achieved by either yeast homologous recombination or In-Fusion HD Cloning Kit (Clontech). A QuikChange Multi Site-Directed Mutagenesis Kit (Agilent Technologies) was used to generate mutation clones of *Scer–APC11*, *Ncas–APC11* and *Klac–APC11*. The substituted codons were optimized for expression in *S. cerevisiae*.

## Genetic procedures of orthologue replacement

The orthologue compatibility test was conducted according to the following procedure (Extended Data Fig. 1). First, the pRS416–Scer plasmid carrying a *S. cerevisiae* essential gene and the *URA3* marker was transformed into the JYL1821 *S. cerevisiae* strain, which was derived from R1158 (*MATa met15-0 ura3::CMV–tTA–ura3 his3-1 leu2-0 trp1-63*). Next, the genomic copy of the essential gene was deleted by transforming PCR-amplified deletion kanMX4 cassettes from the yeast essential heterozygous diploid collection (GE Dharmacon). Successful knockout strains were then transformed with the pRS413–tet–orthologue plasmids. If the orthologous gene could replace the function of the *S. cerevisiae* essential gene, it would allow cells to lose the pRS416–Scer plasmid and grow on 5-Fluoroorotic acid (5-FOA) plates that are toxic to cells with Ura3 activity. At least three biological repeats were performed for each replacement experiment and the results were reported only when consistent patterns were observed. Orthologues were classified as incompatible if all tested colonies could not grow on 5-FOA plates (cell survival rate <64%, binomial test, $P = 0.047$). Inviability of an orthologue-carrying shuffling strain on 5-FOA plates indicates that the essential gene has changed to such an extent that it cannot be replaced by its orthologue.

To generate the *apc2Δ apc11Δ* plasmid shuffling strain, the CDSs and regulatory elements of *Scer–APC2* and *Scer–APC11* were placed in the same pRS416 plasmid and then transformed into the JYL1821 strain. The genomic copies of *Scer–APC2* and *Scer–APC11* were deleted using *kanMX6* and *hphMX4* deletion cassettes, respectively.

To test the compatibility of *Scer–APC2* and *Scer–APC11* in the *N. castellii* background, the CDSs of *Scer–APC2* and *Scer–APC11* were fused with a highly expressed *Ncas–ADH1* promoter and then inserted into the *N. castellii* genome to replace the endogenous *Ncas–APC2* and *Ncas–APC11* genes. All the genetic manipulations were done in a diploid *N. castellii* strain, CBS4310. After the transformants were confirmed by PCR of the insertion sites, the heterozygous replacement lines were induced to sporulation and viable spores were examined. The genetic results indicated that cells carrying only *Scer–APC2* were inviable, but *Scer–APC2 + Scer–APC11* double replacement cells were viable.

## Maintenance and transformation of yeasts

The culture temperature was 28 °C for *S. cerevisiae* and 25 °C for *N. castellii*, *K. latics*, *Y. lipolytica* and *S. pombe*. We used YPD culture medium (1% yeast extract, 2% peptone, 2% glucose) for *S. cerevisiae*, *N. castellii*, *K. latics* and *Y. lipolytica*, whereas YES medium (1% yeast extract, 2% glucose) was used for *S. pombe*.

We used the lithium acetate method to perform transformations in *S. cerevisiae*[56]. For transformations in *N. castellii*, we adopted an electroporation protocol[57] with slight modifications. Briefly, we resuspended 3–5 ml *N. castellii* overnight culture in 8 ml ddH$_2$O, 1 ml 1 M LiOAc, 1 ml 10X TE and 250 μl 1 M DTT, and incubated it for 0.5–2 hr. Then the cell pellet was washed with 30 ml cold ddH$_2$O and then with 5 ml cold 1 M sorbitol. After washing, the cell pellet was resuspended in 0.5 ml cold 1 M sorbitol and kept on ice before transformation. We then mixed 100 μl of cell suspension with 20 μl DNA and put the mixture in a pre-chilled 2 mm electroporation cuvette (BasicLife Bioscience). Electroporation was carried out in an ECM 630 Electroporation System (BTX) with the following settings: 1.8 kV, 200 Ω, 20 μF. After electroporation, 1 ml sorbitol was immediately added to the cell suspension and left to recover for 1 hr at 25 °C before plating on YPD plates.

## RNA isolation and cDNA preparation

We used the Qiagen RNeasy Mini Kit (Qiagen) to isolate RNA from exponentially growing yeasts. The isolated RNA was reverse-transcribed to cDNA by SuperScript III Reverse Transcriptase (Invitrogen). The cDNA was used as templates for PCR and plasmid constructions (Supplementary Table 9).

## Candidate gene selection and sequence analysis

We included the essential genes identified by ref. 1 and the list of essential genes from the *Saccharomyces* Genome Database (https://www.yeastgenome.org) for our analysis. Only genes essential to strain background S288c were considered. We focused on genes with only one orthologue in the other four selected species—*N. castellii*, *K. lactis*, *Y. lipolytica* and *S. pombe*—yielding a total of 796 essential genes.

In the first round of experiments for testing orthologue compatibility, we selected essential genes involved in various cellular functions and also some from the top or bottom ranked genes in terms of sequence conservation because we speculated that these genes might reveal different evolutionary trajectories. We successfully obtained deletion strains and orthologue plasmids for 74 genes. In the second round of experiments, we included another 12 essential genes encoding the subunits of fast-evolving protein complexes to test our hypothesis of co-evolution. Data from both experiments were combined for our analyses.

Identities and similarities between *S. cerevisiae* proteins and those of other yeasts were established in EMBOSS Matcher with default settings[58]. To calculate Ka, protein sequences were first aligned by MUSCLE (version: muscle3.8.31_i86linux64)[59] and then assessed in ParaAT followed with KaKs_Calculator 2.0[60,61].

## Growth rate measurements

Cells of different strains were cultured in 96-well YPD plates at 28 °C until saturation, that is, the cell density of each well was similar to the control strain. Then, saturated cell cultures were diluted 30-fold in fresh YPD and the growth rates were measured using an Infinite 200 (or F200) series plate reader (Tecan). The control strain was always included in each plate, and at least three replicates were measured for each sample.

## Modelling the structure of *S. cerevisiae* APC2–APC11

Currently, there are no available protein structures of *S. cerevisiae* Apc2 or Apc11, so we modelled these yeast protein structures in web-based Phyre2[62]. Among the best templates predicted by Phyre2, we chose the human APC/C structure (Protein Data Bank: 4UI9) as the template for both *S. cerevisiae* Apc2 and Apc11. We used PyMol visualization software (Schrodinger) to show the modelled protein structure.

## Reporting summary

Further information on research design is available in the Nature Portfolio Reporting Summary linked to this article.

## Data availability

Data generated or analysed in this study are provided in the article or its Supplementary Information. Source data are provided with this paper.

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

## Acknowledgements

We thank the members of Leu lab for the helpful discussions and comments on the manuscript. We also thank J. O'Brien for manuscript editing, and K. Swamy and the IMB Bioinformatics Core for the help with bioinformatic analysis. This work was supported by Academia Sinica of Taiwan (grant nos. AS-IA-110-L01 and AS-GCS-110-01) and the National Science and Technology Council of Taiwan (NSTC 111-2326-B-001-015).

## Author contributions

J.-Y.L. conceived the study. H.-Y.L. and J.-Y.L. designed analyses and interpreted results. H.-Y.L., Y.-H.Y. and Y.-T.J. performed the experiments. H.-Y.L. and C.-W.L. performed bioinformatics analysis. H.-Y.L. and J.-Y.L. wrote the paper.

## Competing interests

The authors declare no competing interests.

## Additional information

**Extended data** is available for this paper at https://doi.org/10.1038/s41559-023-02029-5.

**Correspondence and requests for materials** should be addressed to Jun-Yi Leu.

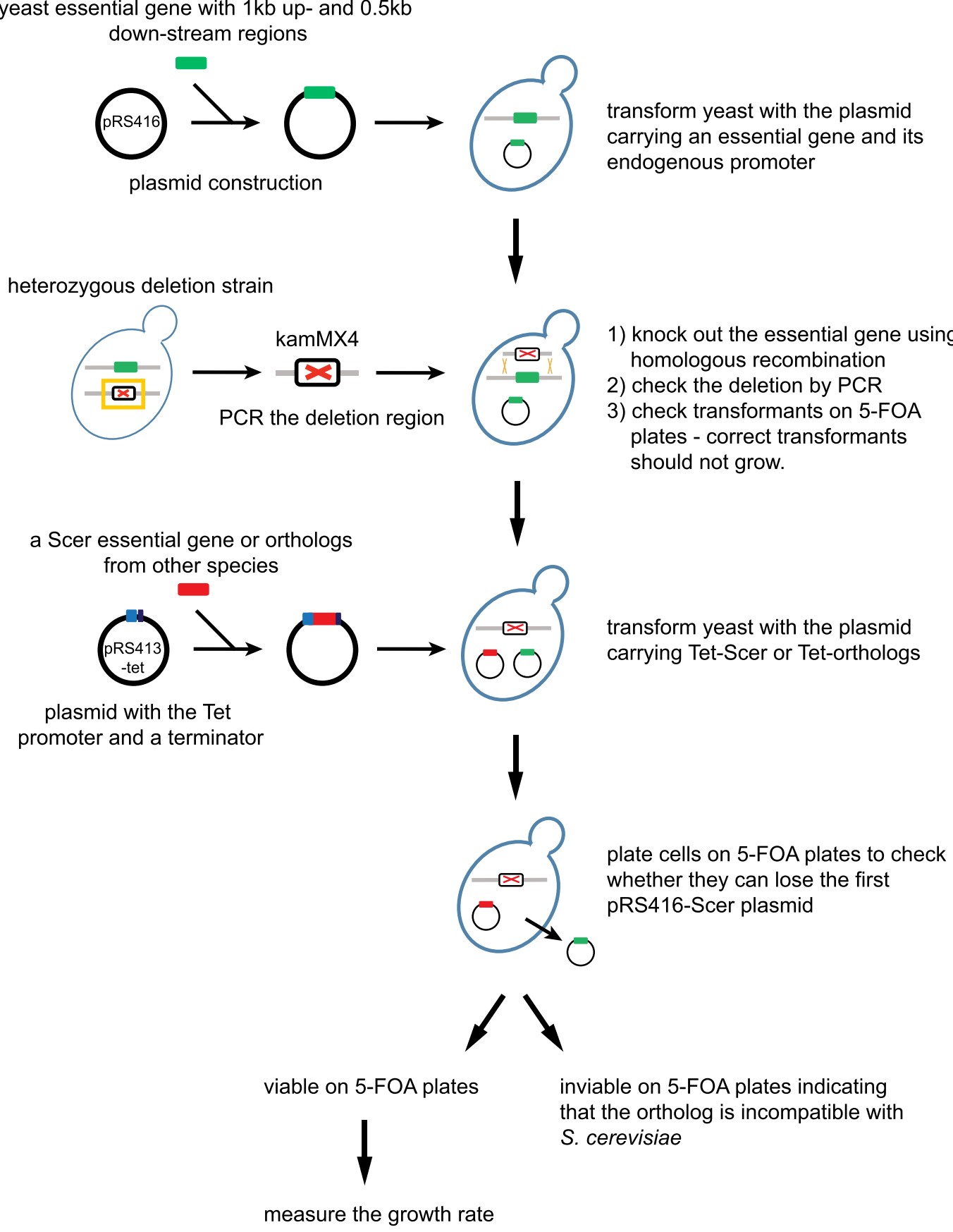

**Extended Data Fig. 1 | Experimental protocol for orthologous gene replacement in *S. cerevisiae*.** The pRS416–Scer plasmid was transformed into a *S. cerevisiae* haploid strain JYL1821 before the essential gene was deleted. A pRS413–tet–Scer or pRS413–tet–orthologue plasmid was then transformed into the deletion strain to determine whether it could replace the pRS416–Scer plasmid. If the essential gene could be replaced by its orthologues, the growth rates of the replacement strains were measured.

**a**

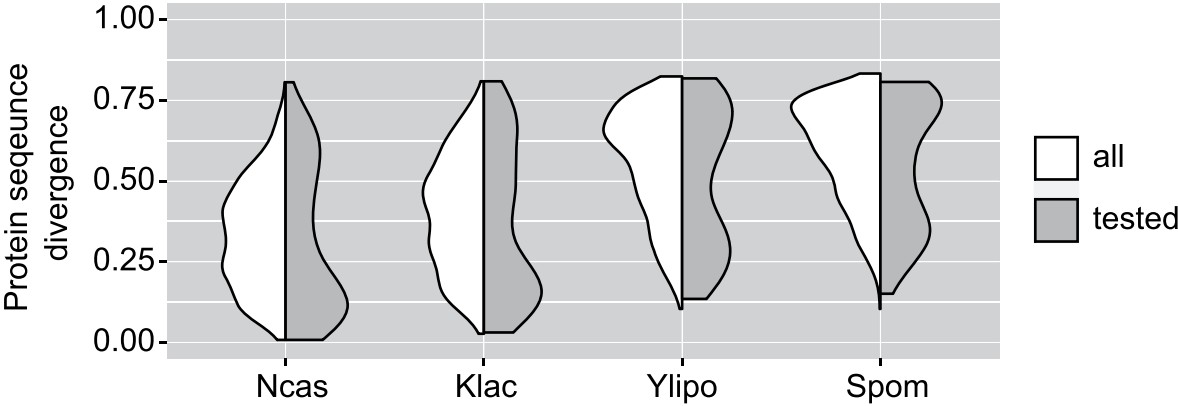

**b**

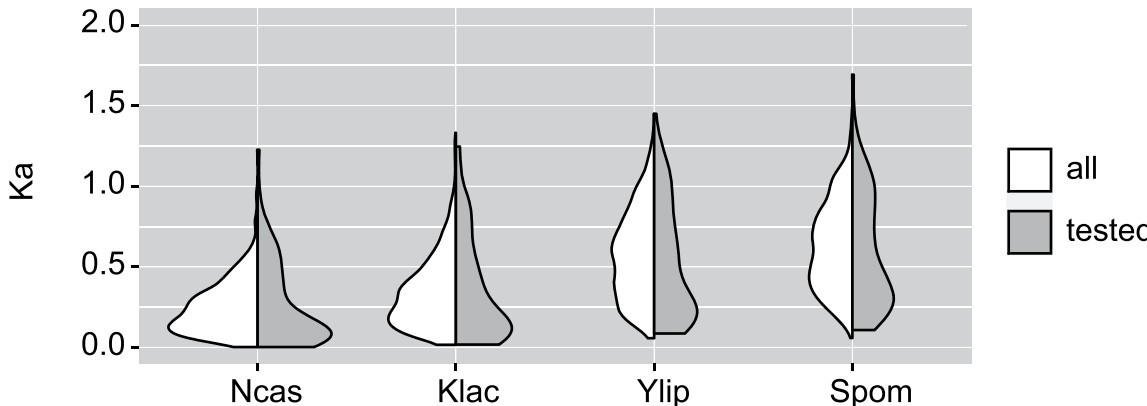

**Extended Data Fig. 2 | Distribution of protein sequence conservation and non-synonymous substitution rates in all essential genes and tested candidates.** Fungal orthologues were compared to the corresponding genes in *S. cerevisiae*. Protein divergence (**a**) was calculated as one minus protein identity, and Ka (**b**) represents the non-synonymous substitution rate. A list of 86 candidate genes (see Supplementary Tables 1 and 2) was selected from the 796 *S. cerevisiae* essential genes with single orthologues in all four tested species. Candidate genes with high and low sequence divergence were slightly overrepresented since we speculated that these two types of genes might reveal specific evolutionary patterns.

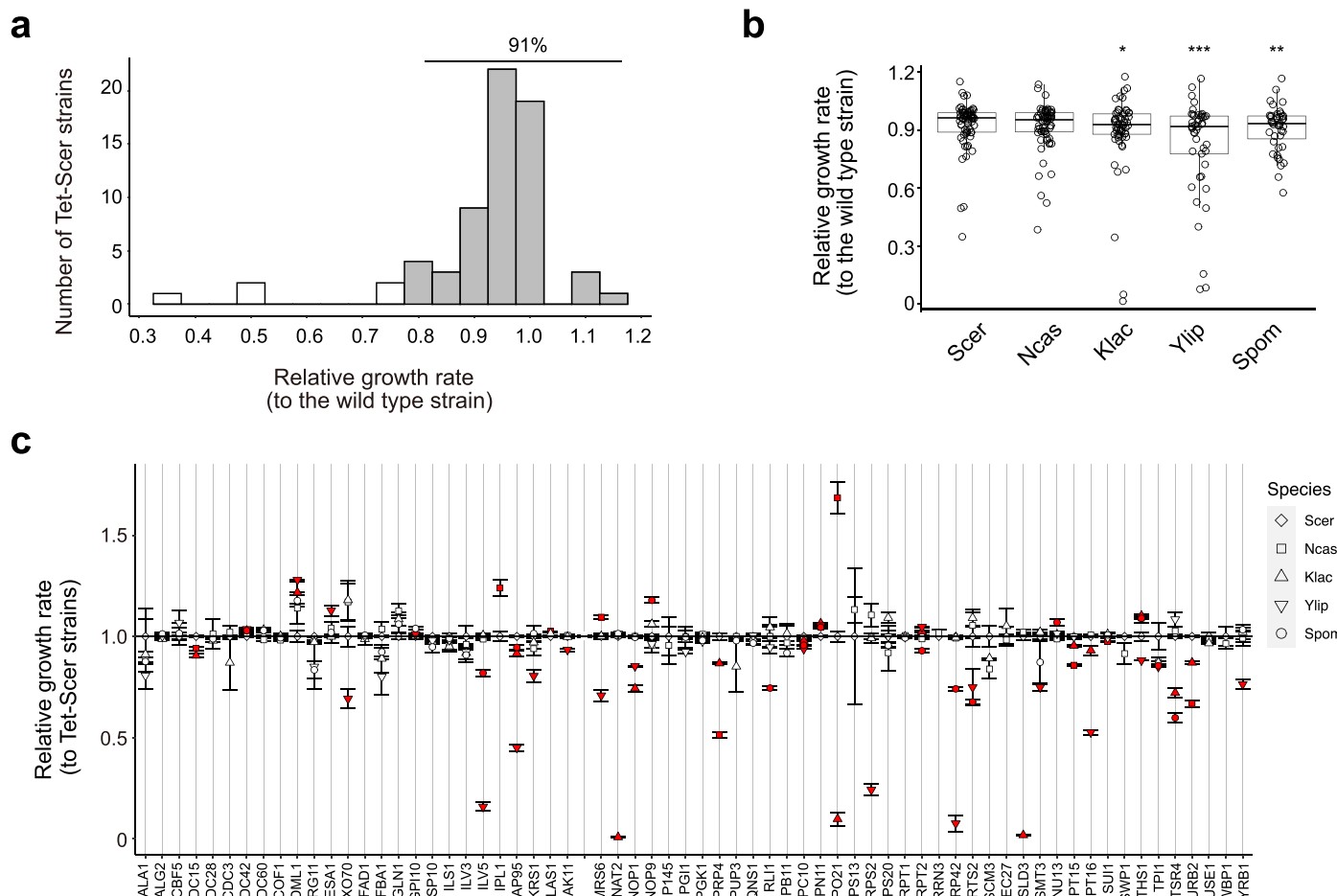

**Extended Data Fig. 3 | Small changes have accumulated in the compatible orthologues. a**, Replacing the endogenous promoter with the Tet-Off promoter does not have a strong impact on the majority of tested genes. The histogram shows the relative growth rates of strains carrying different *S. cerevisiae* essential genes under the Tet-Off promoter. The strains exhibiting less than 20% difference in growth rate are colored grey. **b**, Strains carrying distant orthologues are more likely to show different growth rates. Boxplots indicate median (middle line), 25th and 75th percentile (box), and min and max (whiskers) of relative growth rates in the strains carrying orthologues from different species. Different species data were compared to that of the Tet−Scer strains. *P* values were calculated by paired, two-tailed Student's *t*-test. *, *P* < 0.05. **, *P* < 0.01. ***, *P* < 0.005. The

relative growth rates in (**a**) and (**b**) were obtained by normalizing the growth rate of the Tet promoter strain to the wild-type strain (JYL1821). **c**, Growth rates of the strains carrying different orthologues. To simplify comparisons, all growth rates were normalized to that of individual Tet−Scer strains. ◇Tet−Scer-orthologues; □Tet−Ncas-orthologues; △ Tet−Klac-orthologues; ▽ Tet−Ylip-orthologues; ◯ Tet−Spom−orthologues. Different species data were compared to that of the Tet−Scer strains. Strains significantly different from Tet−Scer strains (two-tailed Dunnett's test, *P* < 0.05) are in red. All strains were grown in rich medium (YPD). Error bars represent the standard error of the mean from three independent colonies (see Supplementary Table 4 for data and detailed statistical information).

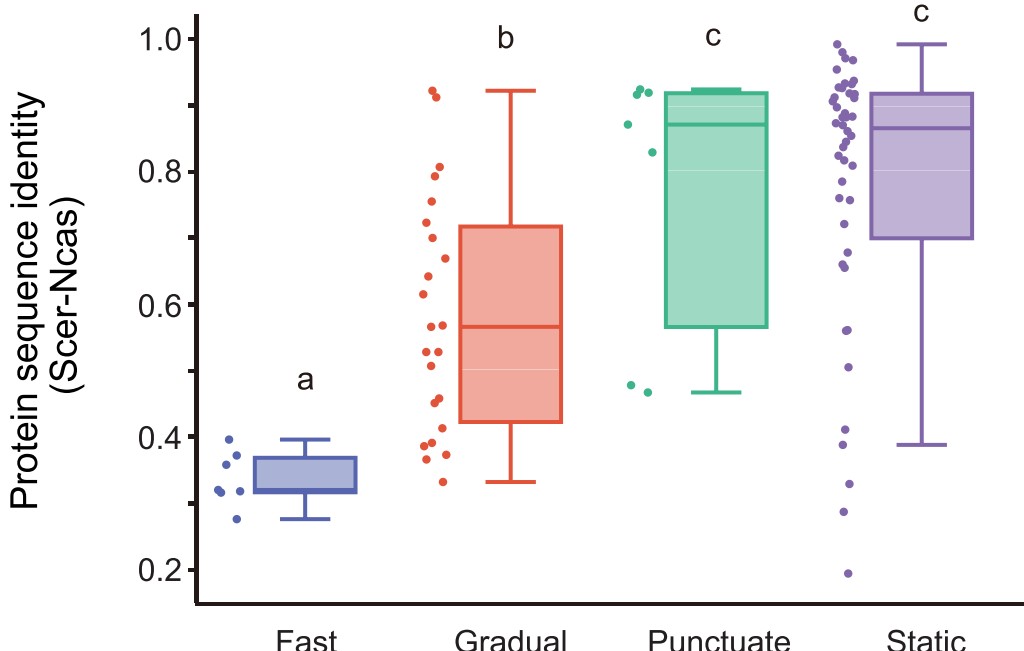

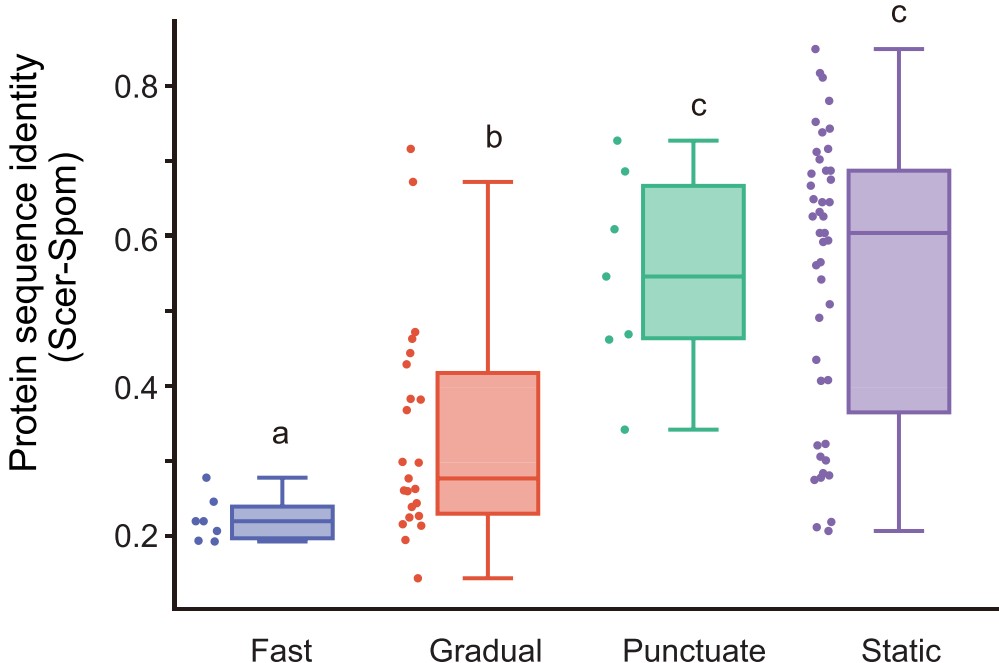

**Extended Data Fig. 4 | The fast-type genes have the lowest protein sequence identity.** Protein identity of each gene was calculated between *S. cerevisiae* and *N. castellii* (**a**) or *S. cerevisiae* and *S. pombe* (**b**), and different groups were compared. Boxplots indicate median (middle line), 25th and 75th percentile (box), and min and max (whiskers). Distributions with different letters (above each boxplot) are significantly different from each other (fast: $n=7$, gradual: $n=24$, puncutate: $n=7$, static: $n=44$, two-sided Mann–Whitney $U$ test, $P$ values < 0.05). Source data and detailed statistical information are provided as a source data file.

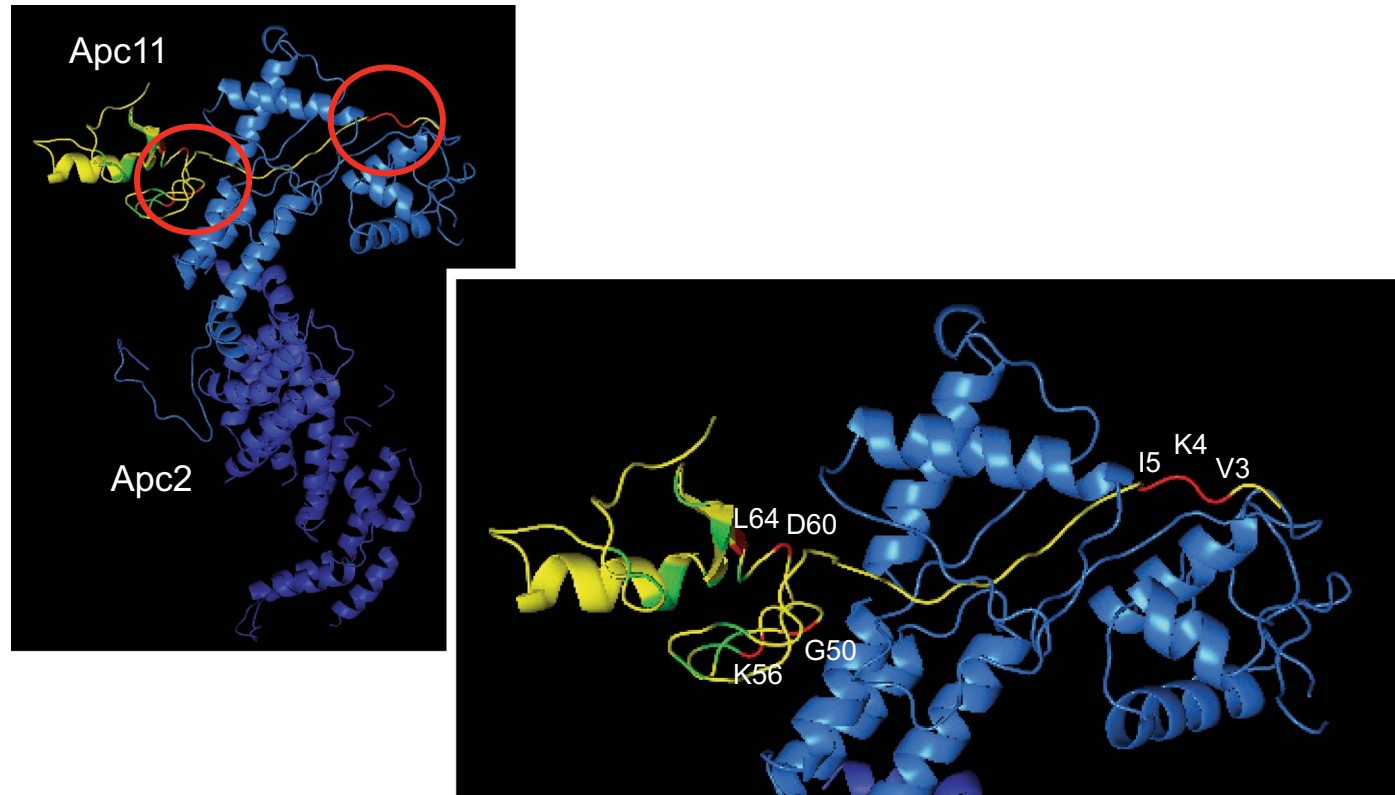

**Extended Data Fig. 5 | Divergent residues of Apc11 are located near the interaction interface of Apc2.** The structures of a complete Scer–Apc2 protein and the first 102 residues of Scer–Apc11 were predicted by the Phyre2 server using the human APC/C complex as the template. The predicted structures of Scer–Apc2 and Scer–Apc11 were then aligned to PDB:4UI9 by the PyMOL Molecular Graphics System (see Methods for details). Previous structural studies have shown that only the C-terminal domain of Scer–Apc2 (residues 430–853, in light blue) interacts with Scer–Apc11, and that the N-terminal domain of Scer–Apc2 (residues 1–419, in dark blue) interacts with other APC/C subunits[27,33]. Scer–Apc11 is colored yellow. The zinc-chelating residues of the canonical RING domain are highlighted in green. The residues chosen for mutagenesis are colored red and numbered.

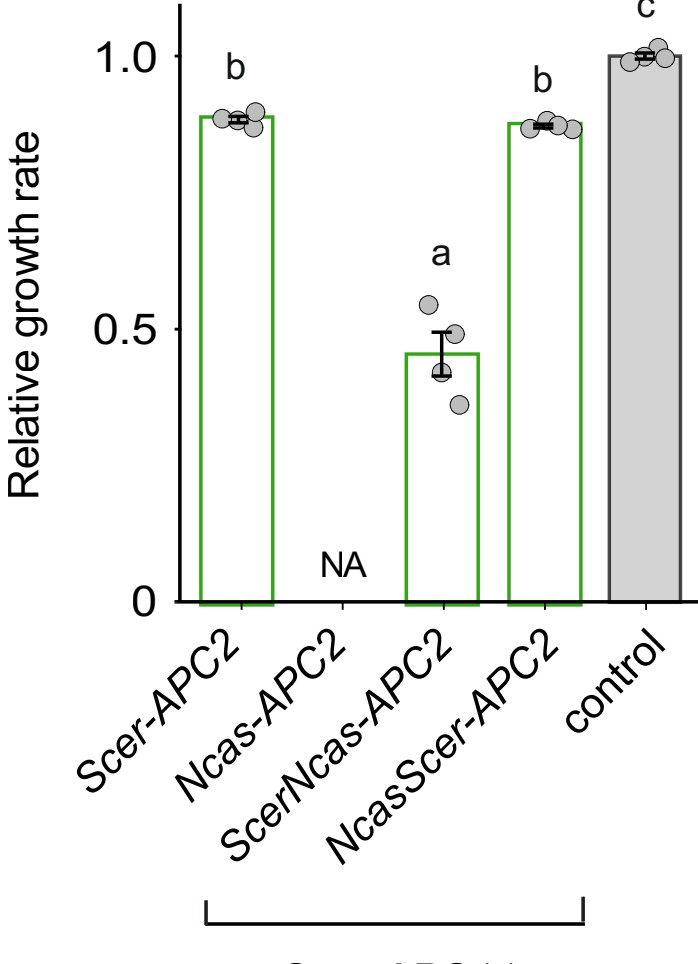

**Extended Data Fig. 6 | Cells show growth defects when the chimeric Apc11 proteins are driven by the endogenous *Scer–APC11* promoter, suggesting that both the CTD and NTD of Apc2 contribute to the stability of the complex.** All compatibility tests of different proteins were performed in an *apc2Δ apc11Δ* shuffling strain with plasmids carrying the orthologous or chimeric genes. The growth rates were normalized to that of a control strain (*apc2Δ apc11Δ* + *Scer–APC2 Scer–APC11*). NA, data not available due to cell inviability. Error bars represent the standard error of the mean from four independent colonies. Columns sharing the same letters are not significantly different (alpha = 0.05, Tukey-adjusted *P* values). Source data and detailed statistical information are provided as a source data file.

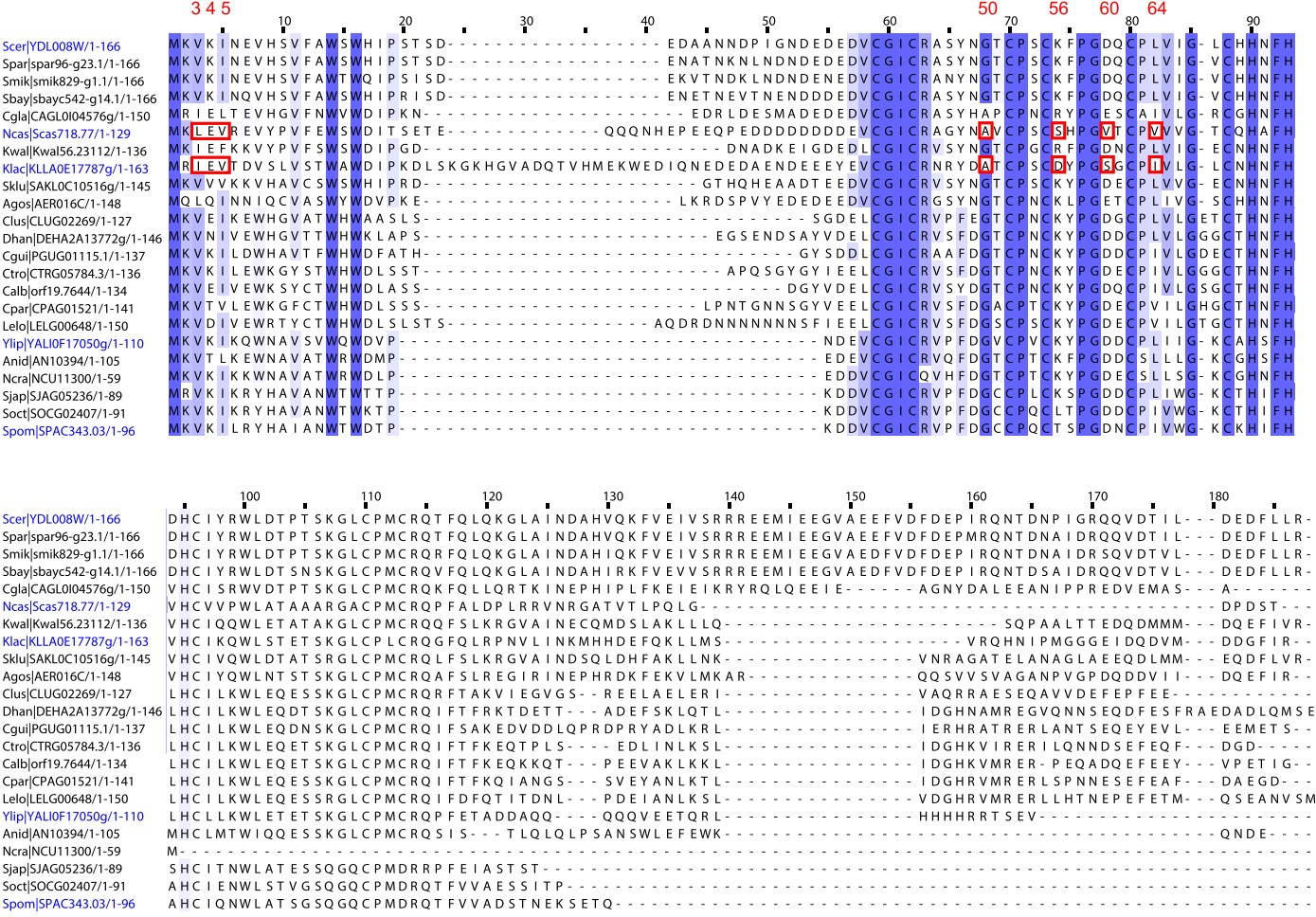

**Extended Data Fig. 7 | Alignment of Apc11 sequences from 23 Ascomycota species reveals a few divergent residues in the highly conserved domains.** The fungal sequences of Apc11 were aligned by MUSCLE[59]. The residues specific to incompatible Ncas–Apc11 and Klac–Apc11 are highlighted in red. The species with tested orthologues are labeled in blue. Residue numbers are based on the Scer–Apc11 sequence.

# Reporting Summary

## Statistics

For all statistical analyses, confirm that the following items are present in the figure legend, table legend, main text, or Methods section.

| n/a | Confirmed | |
|---|---|---|
| ☐ | ☒ | The exact sample size (*n*) for each experimental group/condition, given as a discrete number and unit of measurement |
| ☐ | ☒ | A statement on whether measurements were taken from distinct samples or whether the same sample was measured repeatedly |
| ☐ | ☒ | The statistical test(s) used AND whether they are one- or two-sided<br>*Only common tests should be described solely by name; describe more complex techniques in the Methods section.* |
| ☒ | ☐ | A description of all covariates tested |
| ☒ | ☐ | A description of any assumptions or corrections, such as tests of normality and adjustment for multiple comparisons |
| ☐ | ☒ | A full description of the statistical parameters including central tendency (e.g. means) or other basic estimates (e.g. regression coefficient) AND variation (e.g. standard deviation) or associated estimates of uncertainty (e.g. confidence intervals) |
| ☐ | ☒ | For null hypothesis testing, the test statistic (e.g. *F*, *t*, *r*) with confidence intervals, effect sizes, degrees of freedom and *P* value noted<br>*Give P values as exact values whenever suitable.* |
| ☒ | ☐ | For Bayesian analysis, information on the choice of priors and Markov chain Monte Carlo settings |
| ☒ | ☐ | For hierarchical and complex designs, identification of the appropriate level for tests and full reporting of outcomes |
| ☒ | ☐ | Estimates of effect sizes (e.g. Cohen's *d*, Pearson's *r*), indicating how they were calculated |

*Our web collection on statistics for biologists contains articles on many of the points above.*

## Software and code

Policy information about availability of computer code

| Data collection | No specific data were collected in the current manuscript. |
|---|---|
| Data analysis | Ka of orthologous proteins: sequences were aligned by MUSCLE (version: muscle3.8.31_i86linux64) and then assessed in ParaAT followed with KaKs Calculator 2.0.<br>Apc2-Apc11 interactions were modeled using the web-based Phyre2. |

For manuscripts utilizing custom algorithms or software that are central to the research but not yet described in published literature, software must be made available to editors and reviewers. We strongly encourage code deposition in a community repository (e.g. GitHub). See the Nature Portfolio guidelines for submitting code & software for further information.

## Data

Policy information about availability of data

All manuscripts must include a data availability statement. This statement should provide the following information, where applicable:
- Accession codes, unique identifiers, or web links for publicly available datasets
- A description of any restrictions on data availability
- For clinical datasets or third party data, please ensure that the statement adheres to our policy

All the data were included in the manuscript.

## Human research participants

Policy information about studies involving human research participants and Sex and Gender in Research.

| | |
|---|---|
| Reporting on sex and gender | Not applicable. |
| Population characteristics | Not applicable. |
| Recruitment | Not applicable. |
| Ethics oversight | Not applicable. |

Note that full information on the approval of the study protocol must also be provided in the manuscript.

# Field-specific reporting

Please select the one below that is the best fit for your research. If you are not sure, read the appropriate sections before making your selection.

☒ Life sciences  ☐ Behavioural & social sciences  ☐ Ecological, evolutionary & environmental sciences

For a reference copy of the document with all sections, see nature.com/documents/nr-reporting-summary-flat.pdf

# Life sciences study design

All studies must disclose on these points even when the disclosure is negative.

| | |
|---|---|
| Sample size | Sample size was chosen based on previously published literature on the same subject. Three or more to enable calculation of standard deviation and significance. |
| Data exclusions | No data was excluded. |
| Replication | Reproducibility was confirmed. |
| Randomization | Samples were not randomized, appropriate controls were included in each figure. |
| Blinding | Blinding is not relevant to this study, which does not include any patient or clinical assessments. |

# Reporting for specific materials, systems and methods

We require information from authors about some types of materials, experimental systems and methods used in many studies. Here, indicate whether each material, system or method listed is relevant to your study. If you are not sure if a list item applies to your research, read the appropriate section before selecting a response.

### Materials & experimental systems

| n/a | Involved in the study |
|---|---|
| ☒ ☐ | Antibodies |
| ☒ ☐ | Eukaryotic cell lines |
| ☒ ☐ | Palaeontology and archaeology |
| ☒ ☐ | Animals and other organisms |
| ☒ ☐ | Clinical data |
| ☒ ☐ | Dual use research of concern |

### Methods

| n/a | Involved in the study |
|---|---|
| ☒ ☐ | ChIP-seq |
| ☒ ☐ | Flow cytometry |
| ☒ ☐ | MRI-based neuroimaging |

