## [Peer Review File · Nature Ecology & Evolution]

Peer Review Information

Journal: Nature Ecology & Evolution

Manuscript Title: Multiple intermolecular interactions facilitate rapid evolution of essential genes

Corresponding author name(s): Jun-Yi Leu

Editorial Notes:

Reviewer Comments & Decisions:

Decision Letter, initial version:

10th August 2022

Dear Dr Leu,

Your manuscript entitled "Multiple intermolecular interactions facilitate rapid evolution of essential genes" has now been seen by three reviewers, whose comments are attached. The reviewers have raised a number of concerns which will need to be addressed before we can offer publication in Nature Ecology & Evolution. We will therefore need to see your responses to the criticisms raised and to some editorial concerns, along with a revised manuscript, before we can reach a final decision regarding publication.

We therefore invite you to revise your manuscript taking into account all reviewer and editor comments. Please highlight all changes in the manuscript text file in Microsoft Word format.

* If you have not done so already please begin to revise your manuscript so that it conforms to our Article format instructions at <http://www.nature.com/natecolevol/info/final-submission>. Refer also to any guidelines provided in this letter.

2[REDACTED]

Nature Ecology & Evolution is committed to improving transparency in authorship. As part of our efforts in this direction, we are now requesting that all authors identified as 'corresponding author' on published papers create and link their Open Researcher and Contributor Identifier (ORCID) with their account on the Manuscript Tracking System (MTS), prior to acceptance. ORCID helps the scientific community achieve unambiguous attribution of all scholarly contributions. You can create and link your ORCID from the home page of the MTS by clicking on 'Modify my Springer Nature account'. For more information please visit www.springernature.com/orcid.

[REDACTED]

Reviewer expertise:

Reviewer #1: systems biology, humanization of yeast genes

Reviewer #2: yeast evolutionary genetics

Reviewer #3: fungi evolutionary genetics

Reviewers' comments:

Reviewer #1 (Remarks to the Author):

Summary:

This manuscript presents an interesting study to swap orthologous genes from 4 different and evolutionarily diverse organisms into Baker's yeast. The study has implications for how essential

2orthologs evolve among diverged species. However, there are a few concerns that need to be addressed, which I have listed as the following:

Specific comments:

1. Are the selected set of genes 1:1 orthologs or have duplicates in either lineage? And if there are cases with duplicated genes, the authors should analyze whether the divergent functions may explain their data. Also, could the authors provide the classification of genes based on whether the proteins work alone or are involved in protein-protein interactions?
2. The assumption that genes belonging to gradual-type vs. fast-type are slow and fast evolving genes are not well explained. Do the authors have data showing the slow vs. fast evolution of genes belonging to this class? Or is this statement made from the data obtained from complementation assays alone? Specifically, the follow-up work with fast-type genes harboring single-amino acid change enables complementation. However, the data doesn't reconcile as the fast-evolving genes with many variations require a single amino acid change to rescue incompatibility. Broadly, the complementation assays alone should not serve as an estimate of fast evolution. However, several statements made in this manuscript seem to convey that notion. For example, "How can these essential genes exhibit such rapid changes at the molecular level but still maintain their essential physiological functions?" suggests rapid evolution. Can the authors provide an estimate of this property besides complementation?
3. The statements such as "The "fast" type genes (8/85 = 9%) have become incompatible in species as closely related as *N. castellii* and *S. cerevisiae* (i.e. within 50 million years), indicating that even essential genes can change quickly" Lines 150-152. It is unclear what the word "change" actually means. Does the "change" refer to change of function? If so, have the authors provided evidence?
4. The fascinating part of the manuscript is the data showing the co-expression of interacting partners that enables functional complementation. In the case of pair-wise complementation assays, is the co-expressed protein also an ortholog? And is the co-expressed protein the only interacting partner known based on biochemical assays?
5. The notion that higher order epistasis stabilizes interactions, enabling compatibility is solely based on the growth rate comparisons and seems farfetched.

Reviewer #2 (Remarks to the Author):

In this manuscript Lai et al systematically replace a large number of yeast essential genes with orthologs from other species that are increasingly divergent from *S. cerevisiae*. They identify genes that can be replaced with any of the orthologs, but others that are only rescued with a subset of the orthologs. They called these 'gradual', 'punctate', or 'fast'. Analyzing a subgroup of orthologs that they labeled as 'fast', they find that the APC/C has a complex evolutionary history where incompatibility with the *S. cerevisiae* genome is dependent on a network of interactions with other APC genes. Overall, I really like the experiments, and I think the data produced is very clean. I see no major experimental flaws. I wish maybe the authors would have spent a bit more time analyzing the bulk of the data such as the non-APC genes (are complexes more likely to share the same incompatibility

3profiles? Etc).

What I do not like is the concept of evolutionary pace as described here. Maybe this is simply in need of clarification, or maybe this needs to be reworded. I have a few major comments:

1) I believe that the tempo of evolutionary incompatibility is a bit misleading as is. All but the 'static' case scenario are parsimoniously derived from a single mutational event that leads to incompatibility (or alternatively, that lead to compatibility if the orthologs are mutually incompatible along the tree). The authors present the tempo case as a linear timeline but the events on a phylogenetic tree. In fact, the word description of 'gradual' is pretty much what is seen in all the examples. For some reason the authors describe as time starting from the extant *S. cerevisiae*. It's not clear why a mutational event arising on one branch vs another branch merits a change in description.

To reliably show pace of incompatibility change, the authors have to show that the number of incompatibility that arises are due to different mechanisms in parallel, which as far as I know has not been shown in this manuscript.

2) The APC case is interesting. However, I wonder if the interpretation for the tempo is all that clear. As we now know, yeast underwent a large hybridization event whose parents are likely members of a *K. lactis*-like lineage, and another from a *Z. rouxii*-like lineage. Resolution of the duplicate gene to a single-copy gene can occur through both routes and certainly did not occur instantly. This can make some orthologs have discordant trees with the species tree. I wonder in this case if the APC11 gene from *N. cas* is derived from the *K. lac* lineage, while the APC11 gene of *S. cerevisiae* is derived from the *Z.rox* lineage.

If this is true, then it would explain why the APC has so many more incompatible proteins (when transferred to the *Scer* background). This doesn't detract from the main finding of the result, but it wouldn't really mean that those proteins are 'fast evolving'.

3) I'm confused by the claim of higher-order epistasis having anything to do with protein interactions. As far as I understand, the claims revolve around essentially mutations within a single protein (APC11). While it is true that introduction of some mutations on different APC11 backbones lead to different results, I'm not sure that this must mean that these mutations cause higher order interaction defects with the rest of the APC. Rather, it is possible that those mutations simply do not fold (or whatever) in some backbones. The authors would have to show the specific additional interaction being tweaked for their claim.

4) I'm also confused by how higher-order epistasis accelerates evolution. Maybe they need to clarify what the mean (sequence evolution? Incompatibility emergence?). I believe protein complexes generally show slower sequence evolution but maybe I am wrong here.

I have a few minor questions/concerns:

1) How do the authors confirm that viability on 5-FOA is by plasmid loss? Sometimes mutations in the URA3 marker also show this phenotype. Do the authors add doxycycline to show that the rescue is due specifically to the orthologous gene?

2) The electroporation protocol is listed as 1.8 kV, 200 W, 20 uF. Is wattage actually the parameter controlled or is this a typo? (Usually the listed parameter would be the resistance in parallel).

3) Did the authors verify that the genes they used are actually essential? The list of essential genes from the deletion collection is well known to have some mistakes (even in S288C). In Fig S3, I recognized the gene MET4, which does not seem to be essential.

4) The authors should probably clarify how they establish statistical significance for growth rate measurements. This will clarify some of the confusion in Fig S3c where some points are red but straight up next the *Scer* line, while some points are white by very far from the line.

4Reviewer #3 (Remarks to the Author):

In this manuscript the authors dissect epistatic interactions between species variants in essential yeast genes. Strengths include the technical fireworks (they make and analyze dozens of transgenic allele replacement strains between yeast species up to >400 million years diverged) and insight into mechanisms by which species acquire extensive changes in a protein complex, putatively under neutral drift, while maintaining conserved function. The work could be improved by a clear statement in the intro about the question of interest, and by improvements to the rigor of the data interpretation in the molecular portion of the study. For the latter, below I have spelled out which conclusions I read as inference and which have been established more conclusively by the authors' experiments.

In a first screening section of the work, the authors profile the ability of single essential genes from four species to complement *S. cerevisiae* (Figure 1 and Table S1; more detailed replicate measurements and statistics should be reported here). Next, the authors show that 6/8 cases in which a given species' allele fails to complement can be rescued by that species' allele of another protein partner from the same complex (Table S5; again, reports of replicate measurements and statistics should be in place).

The core of the manuscript centers on in-depth mechanistic study in one such pair, Apc11 and Apc2, between two species, *N. castellii* and *S. cerevisiae*. The approach is to do interspecific domain and amino-acid replacements of constructs of the two focal genes in the *S. cerevisiae* background. The central results are as follows. (1) The authors identify one amino acid in Apc11, residue 60 (at the interface with Apc2), which is sufficient, when replaced from a donor species into the protein allele of a recipient species, for function (at 50-70% wild-type levels) with the donor species' allele of Apc2. This constitutes a rigorous proof of at least part of the mechanism of the incompatibility between species' alleles at the two genes. It also suggests the beginnings of a model of the trajectory for divergence of the protein pair across evolutionary time. (2) The residue 60 effect is abrogated when the N-terminal domain of Apc2 is from *N. castellii* (and the rest of the background is *S. cerevisiae*). It is rigorous to say that the authors have uncovered epistasis between residue 60 of Apc2 and the N-terminal domain of Apc11. However, the authors' interpretation of this in my view goes beyond what is rigorously defensible: they assume that the mechanism of the epistasis hinges on the interface of the N-terminal domain of Apc2 with other APC components, not Apc11, and they thus sell the result as a case of epistasis between Apc2, Apc11, and other proteins (higher-order epistasis). I would not say this has been fully substantiated by the data as written. To support the authors' model more rigorously they could quantify the affinity between Apc2 and other APC subunits in the presence of their interspecies allele replacements. Or alternatively, they could de-emphasize the argument about higher-order epistasis in this section, since the rest of the manuscript is a sizeable contribution to the literature.

The last section of the manuscript describes other incompatibilities in APC between yeast species (supporting data for Figure 4 should be provided), and the lack of any such incompatibilities in the much smaller SCF complex. This is a satisfying wrap-up in that it contrasts the extensive epistatic divergence in APC with constraint in SCF.

5*****END*****

Author Rebuttal to Initial comments

1. Response to Reviewer 1 (Reviewer's comments in bold, responses in red):

Reviewer #1 (Remarks to the Author):

Summary:

This manuscript presents an interesting study to swap orthologous genes from 4 different and evolutionarily diverse organisms into Baker's yeast. The study has implications for how essential orthologs evolve among diverged species. However, there are a few concerns that need to be addressed, which I have listed as the following:

Specific comments:

1.1 Are the selected set of genes 1:1 orthologs or have duplicates in either lineage? And if there are cases with duplicated genes, the authors should analyze whether the divergent functions may explain their data. Also, could the authors provide the classification of genes based on whether the proteins work alone or are involved in protein-protein interactions?

Response: All of our selected genes have only one ortholog in other lineages (page 23, line 581). Therefore, functional divergence is unlikely due to gene duplication. As suggested by the reviewer, we classified our genes based on whether they have interacting partners using the information from *S. cerevisiae*. The data is now added in Supplementary Table 1 and discussed (page 18, line 440). For the static group, only 52% of the genes encode protein complex subunits. In contrast, 85% of the non-static genes encode protein complex subunits (chi-square test, p value = 0.001). Sequence analysis also shows that interacting proteins have lower protein identity and higher K_a than the non-interacting proteins although the differences are not statistically significant. These data suggest that among our tested essential genes, the proteins that work alone change more slowly and are more likely to maintain compatibility between species.

1.2 The assumption that genes belonging to gradual-type vs. fast-type are slow and fast evolving genes are not well explained. Do the authors have data showing the slow vs. fast evolution of genes belonging to this class? Or is this statement made from the data obtained from complementation assays alone? Specifically, the follow-up work with fast-type genes harboring single-amino acid change enables complementation. However, the data doesn't reconcile as the fast-evolving genes with many variations require a single amino acid change to rescue incompatibility. Broadly, the complementation assays alone should not serve as an estimate of fast evolution. However, several statements made in this manuscript seem to convey that notion. For example, "How can these essential genes exhibit such rapid changes at the molecular level but still maintain their essential physiological functions?" suggests rapid evolution. Can the authors provide an estimate of this property besides complementation?

Response: We made the initial fast and slow evolving statements based on both complementation assays and sequence analyses. Other than the complementation patterns, the genes from "fast", "gradual" and "static" groups all have significant differences in their protein identity and nonsynonymous substitution rate (K_a). In the revised manuscript, we added a new paragraph and new figures (page 7, lines 166-182, Figure 1d, 1e, and Supplementary Figure 4) to describe the sequence analysis results. Besides, we have toned down these statements in the revised manuscript.

Although a single amino acid change could rescue incompatibility in the *Ncas-APC11+Scer-APC2* experiment, *N. castellii* is the most closely related species in our experiment. We assume that ortholog divergence is still small between *N. castellii* and *S. cerevisiae*. On the other hand, we also showed that the same single mutation was not sufficient for complementation in the *Klac-APC11+Scer-APC2* experiment (Figure 3c), indicating that further changes have occurred between *K. lactis* and *S. cerevisiae*.

1.3 The statements such as "The "fast" type genes (8/85 = 9%) have become incompatible in species as closely related as *N. castellii* and *S. cerevisiae* (i.e. within 50 million years), indicating that even essential genes can change quickly"

7Lines 150-152. It is unclear what the word "change" actually means. Does the "change" refer to change of function? If so, have the authors provided evidence?

Response: We are sorry about the confusion. Here we mean losing the ability to complement that may result from changes in the function or interaction. However, we observed in the later experiments that the incompatibility is mainly caused by compromised protein interactions. We have rewritten the sentence to make it more clear (page 9, line 195).

1.4 The fascinating part of the manuscript is the data showing the co-expression of interacting partners that enables functional complementation. In the case of pair-wise complementation assays, is the co-expressed protein also an ortholog? And is the co-expressed protein the only interacting partner known based on biochemical assays?

Response: Yes, the co-expressed protein is also an ortholog. From previous biochemical assays, the tested proteins often had multiple interacting partners (please see Supplementary Table S6). In our experiments, we tested all of them individually but only one of them could rescue the incompatibility.

1.5 The notion that higher order epistasis stabilizes interactions, enabling compatibility is solely based on the growth rate comparisons and seems farfetched.

Response: We agree with the reviewer (and other reviewers who also raise a similar issue) that more experiments may be needed to support the higher order epistasis model. Since higher order epistasis is not the major focus of this manuscript, we have removed all related statements in the revised manuscript.

2. Response to Reviewer 2 (Reviewer's comments in bold, responses in red):

Reviewer #2 (Remarks to the Author):

2.1 In this manuscript Lai et al systematically replace a large number of yeast essential genes with orthologs from other species that are increasingly divergent from *S. cerevisiae*. They identify genes that can be replaced with any of the orthologs, but others that are only rescued with a subset of the orthologs. They called these 'gradual', 'punctate', or 'fast'. Analyzing a subgroup of orthologs that they labeled as 'fast', they find that the APC/C has a complex evolutionary history where incompatibility with the *S. cerevisiae* genome is dependent on a network of interactions with other APC genes.

Overall, I really like the experiments, and I think the data produced is very clean. I see no major experimental flaws. I wish maybe the authors would have spent a bit more time analyzing the bulk of the data such as the non-APC genes (are complexes more likely to share the same incompatibility profiles? Etc).

Response: We agree with the reviewer that further analysis in more protein complexes will be an interesting direction. However, it will require many experiments to carefully examine a single complex. It is out of scope of the current manuscript and we will pursue it in the future.

2.2 What I do not like is the concept of evolutionary pace as described here. Maybe this is simply in need of clarification, or maybe this needs to be reworded. I have a few major comments:

1) I believe that the tempo of evolutionary incompatibility is a bit misleading as is. All but the 'static' case scenario are parsimoniously derived from a single mutational event that leads to incompatibility (or alternatively, that lead to compatibility if the orthologs are mutually incompatible along the tree). The authors present the tempo case as a linear timeline but the events on a phylogenetic tree. In fact, the word description of 'gradual' is pretty much what is seen in all the examples. For some reason the authors describe as time starting from the extant *S. cerevisiae*. It's not clear why a mutational event arising on one

9

branch vs another branch merits a change in description.

To reliably show pace of incompatibility change, the authors have to show that the number of incompatibility that arises are due to different mechanisms in parallel, which as far as I know has not been shown in this manuscript.

Response: We thank the reviewer for raising this interesting point. Indeed, our experimental design only revealed the difference between the *S. cerevisiae* genes and their orthologs in other species. When the orthologs from all other species were not compatible with *S. cerevisiae*, it can be a branch-specific change (shown in Fig. A below) or changes in all species (shown in Fig. B). Although we think that a branch-specific change is less likely based on sequence analysis data (see our discussion below), our complementation experiments did not address this issue directly (except for a few genes that are explained in the next paragraph). Therefore, we removed all descriptions related to the evolutionary tempo in the revised manuscript. Nonetheless, we still keep four types of classification (i.e., static, gradual, punctuate and fast) of these genes so it is easier to describe their specific features in later parts of the manuscript. On the other hand, our sequence analysis using K_a of all possible species pairs indicated that the observed K_a patterns are not specific to the *S. cerevisiae* branch (i.e., high K_a genes between *S. cerevisiae* and X also have high K_a between X_i and X_j) even though our experiments only tested this branch. These data suggest that the “fast” group genes do change quickly in all species (supporting Fig. B). In the revised manuscript, we added a new paragraph and new figures (page 7, lines 166-182, Figure 1d, 1e, and Supplementary Figure 4) to describe the sequence analysis results.

Although we did not systematically show that different mutations accumulated in different species to cause the incompatibility, we did provide one such example in the APC experiment. When we co-expressed *N. castellii* Ncas-APC2+Ncas-APC11 and Nacs-APC4+Ncas-APC5, it rescued the *S. cerevisiae* *apc2* and *apc4* mutants, respectively. However, when we co-expressed *K. lactis* Klac-APC2+ Klac-APC11 and Klac-APC4+ Klac-APC5, they could not rescue the *S. cerevisiae* *apc2* and *apc4* mutants (Supplementary Table 7). These results indicated that other than the incompatible mutations existing between *S. cerevisiae* and *N. castellii*, additional mutations have accumulated in *K. lactis*.

2.3 The APC case is interesting. However, I wonder if the interpretation for the tempo is all that clear. As we now know, yeast underwent a large hybridization event whose parents are likely members of a *K. lactis*-like lineage, and another from a *Z. rouxii*-like lineage. Resolution of the duplicate gene to a single-copy gene can occur through both routes and certainly did not occur instantly. This can make some orthologs have discordant trees with the species tree. I wonder in this case if the APC11 gene from *N. cas* is derived from the *K. lac* lineage, while the APC11 gene of *S. cerevisiae* is derived from the *Z.rox* lineage. If this is true, then it would explain why the APC has so many more incompatible proteins (when transferred to the *Scer* background). This doesn't detract from the main finding of the result, but it wouldn't really mean that those proteins are 'fast evolving'.

Response: The reviewer raised an interesting hypothesis here. Although we cannot completely rule out this possibility, several lines of evidence suggest that it may not be the case. First, if *N. castellii* and *S. cerevisiae* keep different copies of the APC genes from the hybrid genome, we expect that flanking regions of the APC genes should have different gene contents since two homeologous chromosomes lost different genes after whole-genome duplication and only one copy of most duplicated genes was left in the current species. However, the YGOB gene synteny data show that similar orthologous gene sets are kept in the regions flanking to APC genes (APC1, APC2, APC4, APC5 and APC11) in *S. cerevisiae* and *N. castellii*, suggesting that both species maintain the same copy of APC genes after WGD. We have discussed it in the revised manuscript (page 19, lines 459-473). Second, our experiments show that Scer-APC2+Scer-APC11 can also complement the *N. castellii* *apc2* mutation (Supplementary Table 7), suggesting that the incompatibility of the APC complex in these two species is mainly driven by pair-wise interactions. Only the changes in APC2 may not be the major force driving the evolution of APC4-APC5 incompatibility.

2.4 I'm confused by the claim of higher-order epistasis having anything to do with protein interactions. As far as I understand, the claims revolve around essentially mutations within a single protein (APC11). While it is true that introduction of some mutations on different APC11 backbones lead to different results, I'm not sure that this must mean that these mutations cause higher order interaction defects with the rest of the APC. Rather, it is possible that those mutations simply do not fold (or whatever) in some backbones. The authors would have to show the specific additional interaction being tweaked for their claim.

Response: We are sorry about the confusion here. In our APC11 experiments, we had evidence showing that the *apc11* mutations in different backbones were fully functional (see Figure 3b for Ncas-APC11-V60D and Figure 3d for Scer-APC11-D60V) if they coexisted with right interacting partners. We also had evidence that the altered interacting partner, NcasScer-APC2, is fully functional if it coexisted with right interacting partners (Figure 2d). Therefore, it is less likely that the observed defects resulted from the mutant proteins themselves. The definition of higher-order epistasis is that pairwise epistatic interactions themselves can vary with genomic background ¹. Here we saw the same pairwise interaction (Apc11 and the CTD of Apc2) varied its complementation ability depending on another interaction in the APC/C complex (i.e.,

12the interaction between the NTD of Apc2 and other Apc components). These genetic data support the idea of higher-order epistasis. However, since higher order epistasis is not the main focus of the current manuscript, we have removed all higher-order epistasis-related statements in the revised manuscript and emphasized that the effect of mutations is background-dependent.

- 1 Weinreich, D. M., Lan, Y., Jaffe, J. & Heckendorn, R. B. The Influence of Higher-Order Epistasis on Biological Fitness Landscape Topography. *J Stat Phys* **172**, 208-225, doi:10.1007/s10955-018-1975-3 (2018).

2.5 I'm also confused by how higher-order epistasis accelerates evolution. Maybe they need to clarify what the mean (sequence evolution? Incompatibility emergence?). I believe protein complexes generally show slower sequence evolution but maybe I am wrong here.

Response: As discussed in our reply to Reviewer 1 (point 1.2), we observed consistent patterns in incompatibility emergence and sequence change, suggesting that these two features follow the same trend (see also a new paragraph on page 7 lines 166-182, and three new figures, Figure 1d, 1e, and Supplementary Figure 4). The reviewer is correct that protein complexes generally show slower sequence evolution compared to singletons if the whole genome is analyzed together. However, we think that the conclusion (that protein complexes generally show slower sequence evolution) is oversimplified and likely biased for the following reasons. First, complex subunits contain more essential genes (38%, using data from Benschop et al., 2010) than singletons (10%). If only essential genes were analyzed, no significant difference was observed between complex subunits and singletons in yeast. Second, a higher proportion of non-complex proteins belong to the static group than complex subunits in our experiments (page 18, line 440 and also our reply to Reviewer 1, point 1.1). Lastly, we observed that the “fast” group genes all belong to large protein complexes (top 16% in complex size ranking) with multiple interactions (Supplementary Table 6). These data led us to propose the model that higher-order epistasis accelerates evolution (Figure 3h). Nonetheless, we have removed all statements related to higher order epistasis and used multiple interactions to explain our observation.

2.6 I have a few minor questions/concerns:

1) How do the authors confirm that viability on 5-FOA is by plasmid loss? Sometimes mutations in the URA3 marker also show this phenotype. Do the authors add doxycycline to show that the rescue is due specifically to the orthologous gene?

Response: We checked the ura⁻ colonies by PCR the *S. cerevisiae* genes if we found very few colonies on the 5-FOA plates in multiple rounds of the same ortholog replacement experiment.

In most replacement lines, their growth was inhibited by doxycycline. However, some of them were not completely inhibited probably due to only a small amount of gene products was required for cell survival and the Tet promoter was not 100% tight.

2.7 The electroporation protocol is listed as 1.8 kV, 200 W, 20 uF. Is wattage actually the parameter controlled or is this a typo? (Usually the listed parameter would be the resistance in parallel).

Response: Yes, it is a typo of ohm (due to the font change).

2.8 Did the authors verify that the genes they used are actually essential? The list of essential genes from the deletion collection is well known to have some mistakes (even in S288C). In Fig S3, I recognized the gene MET4, which does not seem to be essential.

Response: We selected the genes for testing from the SGD essential gene list. After seeing the reviewer's comment, we went back to check our met4 deletion clones. For some unknown reasons, the mutant clones (carrying pRS416-Scer-MET4) had difficulty losing the plasmid and only very few colonies grew on 5-FOA plates (so we mistook it as an essential genes during the strain construction). However, when we checked by PCR, all the 5-FOA colonies did not carry the plasmid that agreed with the reviewer's

14comment. Since the data suggest that MET4 is not essential, we have removed all MET4-related results from our manuscript.

2.9 The authors should probably clarify how they establish statistical significance for growth rate measurements. This will clarify some of the confusion in Fig S3c where some points are red but straight up next the Scer line, while some points are white by very far from the line.

Response: We performed Dunnett's test for statistical significance. When the points are far from the line and white, they have high standard errors (often due to cell aggregation problems during growth assays since some mutant cells flocculate easily), and vice versa. In the revised manuscript, we added standard errors to the figure to make it more clear.

3. Response to Reviewer 3 (Reviewer's comments in bold, responses in red):

Reviewer #3 (Remarks to the Author):

3.1 In this manuscript the authors dissect epistatic interactions between species variants in essential yeast genes. Strengths include the technical fireworks (they make and analyze dozens of transgenic allele replacement strains between yeast species up to >400 million years diverged) and insight into mechanisms by which species acquire extensive changes in a protein complex, putatively under neutral drift, while maintaining conserved function. The work could be improved by a clear statement in the intro about the question of interest, and by improvements to the rigor of the data interpretation in the molecular portion of the study. For the latter, below I have spelled out which conclusions I read as inference and which have been established more conclusively by the authors' experiments.

Response: We thank the reviewer for the suggestion. We rewrote the introduction to explain the major questions more clearly (page 3, line 56; page 4, lines 68-72, line 89).3.2 In a first screening section of the work, the authors profile the ability of single essential genes from four species to complement *S. cerevisiae* (Figure 1 and Table S1; more detailed replicate measurements and statistics should be reported here). Next, the authors show that 6/8 cases in which a given species' allele fails to complement can be rescued by that species' allele of another protein partner from the same complex (Table S5; again, reports of replicate measurements and statistics should be in place).

Response: In the revised manuscript, we reported the biological repeat number for the data shown in Figure 1, Table S1 and S5 (page 31, line 821). Since the complementation assay results were reported only when 3 individual biological repeats exhibited the same phenotype (i.e., cells carrying the ortholog could survive on 5-FOA plates or not), no statistical test is required for these data. We also performed the chi-square test to show that incompatible gene numbers in different species were significantly different (page 31, line 826).

3.3 The core of the manuscript centers on in-depth mechanistic study in one such pair, *Apc11* and *Apc2*, between two species, *N. castellii* and *S. cerevisiae*. The approach is to do interspecific domain and amino-acid replacements of constructs of the two focal genes in the *S. cerevisiae* background. The central results are as follows. (1) The authors identify one amino acid in *Apc11*, residue 60 (at the interface with *Apc2*), which is sufficient, when replaced from a donor species into the protein allele of a recipient species, for function (at 50-70% wild-type levels) with the donor species' allele of *Apc2*. This constitutes a rigorous proof of at least part of the mechanism of the incompatibility between species' alleles at the two genes. It also suggests the beginnings of a model of the trajectory for divergence of the protein pair across evolutionary time. (2) The residue 60 effect is abrogated when the N-terminal domain of *Apc2* is from *N. castellii* (and the rest of the background is *S. cerevisiae*). It is rigorous to say that the authors have uncovered epistasis between residue 60 of *Apc2* and the N-terminal domain of *Apc11*. However, the authors' interpretation of this in my view goes beyond what is rigorously defensible: they assume that the mechanism of the epistasis hinges on the interface of the N-terminal domain of *Apc2* with other APC components, not *Apc11*, and they thus sell the result as a case of epistasis between *Apc2*, *Apc11*, and other proteins (higher-order epistasis). I would not say

16this has been fully substantiated by the data as written. To support the authors' model more rigorously they could quantify the affinity between Apc2 and other APC subunits in the presence of their interspecies allele replacements. Or alternatively, they could de-emphasize the argument about higher-order epistasis in this section, since the rest of the manuscript is a sizeable contribution to the literature.

Response: We assume that the contribution of the NTD of Apc2 comes from its interaction with other APC components, not Apc11, for the following reasons. First, previous cryo-EM data have shown that the NTD of Apc2 locates far away from Apc11 in the complex and directly interacts with other APC subunits². Second, in the chimeric Apc2 experiments, we show that ScerNcas-Apc2 (in which the NTD is derived from Scer-Apc2) has higher fitness than Ncas-Apc2 when the partner is Ncas-Apc11 (Figure 2c). If the effect of the NTD comes from the interaction with Apc11, it will be difficult to explain why the NTD of Scer-Apc2 improves the interaction with Ncas-Apc11. However, we also agree with the reviewer that more experiments may be needed to support the higher order epistasis model. Since higher order epistasis is not the major focus of this manuscript, we have removed all the related statements and discussed only interactions between different domains of the proteins.

- 2 Chang, L., Zhang, Z., Yang, J., McLaughlin, S. H. & Barford, D. Molecular architecture and mechanism of the anaphase-promoting complex. *Nature* **513**, 388-393, doi:10.1038/nature13543 (2014).

3.4 The last section of the manuscript describes other incompatibilities in APC between yeast species (supporting data for Figure 4 should be provided), and the lack of any such incompatibilities in the much smaller SCF complex. This is a satisfying wrap-up in that it contrasts the extensive epistatic divergence in APC with constraint in SCF.

We thank the reviewer for the support. The supporting data for Figure 4 are included in Supplementary Table 1. This information has been added.

Decision Letter, first revision:

4th January 2023

Dear Jun-Yi,

Thank you for submitting your revised manuscript "Multiple intermolecular interactions facilitate rapid evolution of essential genes" (NATECOLEVOL-220616682A). It has now been seen again by two of the original reviewers and their comments are below. The reviewers find that the paper has improved in revision, and therefore we'll be happy in principle to publish it in Nature Ecology & Evolution, pending minor revisions to satisfy the reviewers' final requests and to comply with our editorial and formatting guidelines.

I should stress that you will need to include statistics for all analyses as indicated by Reviewer #3.

[REDACTED]

Reviewer #2 (Remarks to the Author):

The authors have addressed all my major concerns in this revision.

There are a few things here and there that may be addressed for precision:

- 1) The abstract still lists 87 genes (as opposed to 86).
- 2) It should be indicated what the authors do for 'Gradual' and 'Punctate' when incompatibility with *S. pombe* is the only observed incompatibility (maybe this is said but I am not sure).
- 3) I think the authors could make it clearer in the figure that K_a was measured between Scer and Ncas, because there's no real reason the 'Punctate' group should evolve slower than the 'Fast' group,

18except if we concentrate on the Scer lineage in this calculation (indeed the 'Fast' group is the same as the 'Punctate' group with a different focal lineage). I see that the authors made the analysis of Scer to X, but they haven't done the 'punctate' to X, which should show the same rate as the 'Fast' group.

Personally, I still do not like the terms 'fast', 'gradual', 'punctate', and 'static' (for many reasons and the reason above). These terms do have real meaning and I'm not convinced they are best used here. However, since it's difficult for me to come up with an alternative set of words, I guess this is fine though perhaps something along the lines of 'X lineage specific' would have been preferable.

Reviewer #3 (Remarks to the Author):

My only remaining comment is that statistics should be in place for all analyses. This includes asterisks for significance in growth rate experiments in Figures 2 and 3, and tests of the compatibility experiments in Figures 1 and 4. The authors' rebuttal for the previous round of review asserts that the latter aren't needed but I don't agree. Qualitative data ("growth" or "no growth") can be tested e.g. with a binomial.

Our ref: NATECOLEVOL-220616682A

16th January 2023

Dear Dr. Leu,

Thank you for your patience as we've prepared the guidelines for final submission of your Nature Ecology & Evolution manuscript, "Multiple intermolecular interactions facilitate rapid evolution of essential genes" (NATECOLEVOL-220616682A). Please carefully follow the step-by-step instructions provided in the attached file, and add a response in each row of the table to indicate the changes that you have made. Please also check and comment on any additional marked-up edits we have proposed within the text. Ensuring that each point is addressed will help to ensure that your revised manuscript can be swiftly handed over to our production team.

****We would like to start working on your revised paper, with all of the requested files and forms, as soon as possible (preferably within two weeks). Please get in contact with us immediately if you anticipate it taking more than two weeks to submit these revised files.****

19If you have not done so already, please alert us to any related manuscripts from your group that are under consideration or in press at other journals, or are being written up for submission to other journals (see: <https://www.nature.com/nature-research/editorial-policies/plagiarism#policy-on-duplicate-publication> for details).

In recognition of the time and expertise our reviewers provide to Nature Ecology & Evolution's editorial process, we would like to formally acknowledge their contribution to the external peer review of your manuscript entitled "Multiple intermolecular interactions facilitate rapid evolution of essential genes". For those reviewers who give their assent, we will be publishing their names alongside the published article.

Nature Ecology & Evolution offers a Transparent Peer Review option for new original research manuscripts submitted after December 1st, 2019. As part of this initiative, we encourage our authors to support increased transparency into the peer review process by agreeing to have the reviewer comments, author rebuttal letters, and editorial decision letters published as a Supplementary item. When you submit your final files please clearly state in your cover letter whether or not you would like to participate in this initiative. Please note that failure to state your preference will result in delays in accepting your manuscript for publication.

Cover suggestions

As you prepare your final files we encourage you to consider whether you have any images or illustrations that may be appropriate for use on the cover of Nature Ecology & Evolution.

Nature Ecology & Evolution has now transitioned to a unified Rights Collection system which will allow our Author Services team to quickly and easily collect the rights and permissions required to publish your work. Approximately 10 days after your paper is formally accepted, you will receive an email in providing you with a link to complete the grant of rights. If your paper is eligible for Open Access, our Author Services team will also be in touch regarding any additional information that may be required to arrange payment for your article.

Please note that *Nature Ecology & Evolution* is a Transformative Journal (TJ). Authors may publish their research with us through the traditional subscription access route or make their paper immediately open access through payment of an article-processing charge (APC). Authors will not be required to make a final decision about access to their article until it has been accepted. [Find out more about Transformative Journals](https://www.springernature.com/gp/open-research/transformative-journals)

Authors may need to take specific actions to achieve [compliance with funder and institutional open access mandates](https://www.springernature.com/gp/open-research/funding/policy-compliance-faqs). If your research is supported by a funder that requires immediate open access (e.g. according to [Plan S principles](https://www.springernature.com/gp/open-research/plan-s-compliance)) then you should select the gold OA route, and we will direct you to the compliant route where possible. For authors selecting the subscription publication route, the journal's standard licensing terms will need to be accepted, including [self-archiving-and-license-to-publish](https://www.nature.com/nature-portfolio/editorial-policies/self-archiving-and-license-to-publish). Those licensing terms will supersede any other terms that the author or any third party may assert apply to any version of the manuscript.

[REDACTED]

[REDACTED]

Reviewer #2:

Remarks to the Author:

The authors have addressed all my major concerns in this revision.

There are a few things here and there that may be addressed for precision:

1) The abstract still lists 87 genes (as opposed to 86).

2) It should be indicated what the authors do for 'Gradual' and 'Punctate' when incompatibility with *S. pombe* is the only observed incompatibility (maybe this is said but I am not sure).

3) I think the authors could make it clearer in the figure that K_a was measured between Scer and Ncas, because there's no real reason the 'Punctate' group should evolve slower than the 'Fast' group, except if we concentrate on the Scer lineage in this calculation (indeed the 'Fast' group is the same as the 'Punctate' group with a different focal lineage). I see that the authors made the analysis of Scer to X, but they haven't done the 'punctate' to X, which should show the same rate as the 'Fast' group.

Personally, I still do not like the terms 'fast', 'gradual', 'punctate', and 'static' (for many reasons and the reason above). These terms do have real meaning and I'm not convinced they are best used here. However, since it's difficult for me to come up with an alternative set of words, I guess this is fine though perhaps something along the lines of 'X lineage specific' would have been preferable.

Reviewer #3:

Remarks to the Author:

My only remaining comment is that statistics should be in place for all analyses. This includes asterisks for significance in growth rate experiments in Figures 2 and 3, and tests of the compatibility experiments in Figures 1 and 4. The authors' rebuttal for the previous round of review asserts that the latter aren't needed but I don't agree. Qualitative data ("growth" or "no growth") can be tested e.g. with a binomial.

Author Rebuttal, first revision:

1. Response to Reviewer 2 (Reviewer's comments in bold, responses in red):

Reviewer #2 (Remarks to the Author):

The authors have addressed all my major concerns in this revision.

There are a few things here and there that may be addressed for precision:

1) The abstract still lists 87 genes (as opposed to 86).

It is corrected.

2) It should be indicated what the authors do for 'Gradual' and 'Punctate' when

incompatibility with *S. pombe* is the only observed incompatibility (maybe this is said but I am not sure).

The information is added (page 7, line 145).

3) I think the authors could make it clearer in the figure that K_a was measured between Scer and Ncas, because there's no real reason the 'Punctate' group should evolve slower than the 'Fast' group, except if we concentrate on the Scer lineage in this calculation (indeed the 'Fast' group is the same as the 'Punctate' group with a different focal lineage). I see that the authors made the analysis of Scer to X, but they haven't done the 'punctate' to X, which should show the same rate as the 'Fast' group.

Figure 1d is modified to include this information that K_a was measured between Scer and Ncas.

Personally, I still do not like the terms 'fast', 'gradual', 'punctate', and 'static' (for many reasons and the reason above). These terms do have real meaning and I'm not convinced they are best used here. However, since it's difficult for me to come up with an alternative set of words, I guess this is fine though perhaps something along the lines of 'X lineage specific' would have been preferable.

We have added this description in Line 142 (page 7).

2. Response to Reviewer 3 (Reviewer's comments in bold, responses in red):

Reviewer #3 (Remarks to the Author):

My only remaining comment is that statistics should be in place for all analyses. This includes asterisks for significance in growth rate experiments in Figures 2 and 3, and tests of the compatibility experiments in Figures 1 and 4. The authors' rebuttal for the previous round of

23review asserts that the latter aren't needed but I don't agree. Qualitative data ("growth" or "no growth") can be tested e.g. with a binomial.

We have modified Figures 2 and 3 to include all the statistical analysis results. For the compatibility experiments in Figure 1 and 4, we have performed binomial tests as the reviewer suggested. The information is described in Line 529-533 (page 21).

Final Decision Letter:

21st February 2023

Dear Jun-Yi,

We are pleased to inform you that your Article entitled "Multiple intermolecular interactions facilitate rapid evolution of essential genes", has now been accepted for publication in Nature Ecology & Evolution.

Over the next few weeks, your paper will be copyedited to ensure that it conforms to Nature Ecology and Evolution style. Once your paper is typeset, you will receive an email with a link to choose the appropriate publishing options for your paper and our Author Services team will be in touch regarding any additional information that may be required

You will not receive your proofs until the publishing agreement has been received through our system

Due to the importance of these deadlines, we ask you please us know now whether you will be difficult to contact over the next month. If this is the case, we ask you provide us with the contact information (email, phone and fax) of someone who will be able to check the proofs on your behalf, and who will be available to address any last-minute problems . Once your paper has been scheduled for online publication, the Nature press office will be in touch to confirm the details.

Acceptance of your manuscript is conditional on all authors' agreement with our publication policies (see www.nature.com/authors/policies/index.html). In particular your manuscript must not be published elsewhere and there must be no announcement of the work to any media outlet until the publication date (the day on which it is uploaded onto our web site).

Please note that *Nature Ecology & Evolution* is a Transformative Journal (TJ). Authors may

24publish their research with us through the traditional subscription access route or make their paper immediately open access through payment of an article-processing charge (APC). Authors will not be required to make a final decision about access to their article until it has been accepted. [Find out more about Transformative Journals](https://www.springernature.com/gp/open-research/transformative-journals)

Authors may need to take specific actions to achieve [compliance with funder and institutional open access mandates](https://www.springernature.com/gp/open-research/funding/policy-compliance-faqs). If your research is supported by a funder that requires immediate open access (e.g. according to [Plan S principles](https://www.springernature.com/gp/open-research/plan-s-compliance)) then you should select the gold OA route, and we will direct you to the compliant route where possible. For authors selecting the subscription publication route, the journal's standard licensing terms will need to be accepted, including [self-archiving-and-license-to-publish](https://www.nature.com/nature-portfolio/editorial-policies/self-archiving-and-license-to-publish). Those licensing terms will supersede any other terms that the author or any third party may assert apply to any version of the manuscript.

We welcome the submission of potential cover material (including a short caption of around 40 words) related to your manuscript; suggestions should be sent to Nature Ecology & Evolution as electronic files (the image should be 300 dpi at 210 x 297 mm in either TIFF or JPEG format). Please note that such pictures should be selected more for their aesthetic appeal than for their scientific content, and that colour images work better than black and white or grayscale images. Please do not try to design a cover with the Nature Ecology & Evolution logo etc., and please do not submit composites of images related to your work. I am sure you will understand that we cannot make any promise as to whether any of your suggestions might be selected for the cover of the journal.

You can generate the link yourself when you receive your article DOI by entering it here: <http://authors.springernature.com/share>.

[REDACTED]

P.S. Click on the following link if you would like to recommend Nature Ecology & Evolution to your librarian <http://www.nature.com/subscriptions/recommend.html#forms>

** Visit the Springer Nature Editorial and Publishing website at http://editorial-jobs.springernature.com?utm_source=ejp_NEcoE_email&utm_medium=ejp_NEcoE_email&utm_campaign=ejp_NEcoE for more information about our career opportunities. If you have any questions please click [here](mailto:editorial.publishing.jobs@springernature.com). **